# Impact of Precipitation Mass Sinks on Midlatitude Storms in Idealized GCM Simulations over a Wide Range of Climates

Tristan H. Abbott[1] and Paul A. O'Gorman[2]

[1]Program in Atmospheric and Oceanic Sciences, Princeton University, Princeton, NJ 08540, USA
[2]Department of Earth, Atmospheric and Planetary Sciences, Massachusetts Institute of Technology, Cambridge, MA 02139, USA

**Correspondence:** Tristan H. Abbott (tristana@princeton.edu)

**Abstract.** Precipitation formation and fallout affects atmospheric flows through the release of latent heat and through the removal of mass from the atmosphere, but because the mass of water vapor is only a small fraction of the total mass of Earth's atmosphere, precipitation mass sinks are often neglected in theory and models. However, a small number of modeling studies suggest that water mass sources and sinks can intensify heavily-precipitating weather systems. These studies point to a need to more systematically verify the impact of neglecting precipitation mass sinks, particularly for warmer and moister climates in which precipitation rates can be much higher. In this paper, we add precipitation mass sources and sinks to an idealized general circulation model and examine their effects on steady-state midlatitude storm track statistics. The model has several idealizations, including that all condensates immediately fall out of the atmosphere, and is run across a wide range of climates, including very warm climates. We find that modifying the model to include mass sources and sinks has no detectable effect on midlatitude variability or extremes, even in climates much warmer and moister than the modern. However, we find that a tenfold exaggeration of mass sources and sinks is sufficient to produce more intense midlatitude weather extremes and increase surface pressure variance. This result is consistent with theoretical potential vorticity analysis, which suggests that the dynamical effects of mass sources and sinks are much smaller than the dynamical effects of accompanying latent heating unless mass sinks are artificially amplified by at least a factor of 10. Finally, we use simulations of "tropical cyclone worlds" to attempt to reconcile our results with earlier work showing stronger deepening in a simulation of a tropical cyclone case study when precipitation mass sinks were included. We demonstrate that abruptly "turning on" mass sources and sinks can lead to stronger transient deepening in some individual storms (consistent with results of past work) but weaker transient deepening in other storms, without modifying the steady-state statistics of storms in equilibrium with the large-scale environment (consistent with our other results). Our results provide a firmer foundation for using general circulation models that neglect moist mass sources and sinks in climate simulations, even in climates much warmer than today, while leaving open the possibility that their inclusion might lead to short-term improvements in forecast skill.

## 1 Introduction

In midlatitudes, large-scale atmospheric flows are dominated by extratropical cyclones and anticyclones. These disturbances play an important role in climate and the general circulation of the atmosphere through transports of heat, moisture, and angular

momentum (Held, 1975; Pierrehumbert and Swanson, 1995; Chang et al., 2002). Additionally, they organize extratropical precipitation (Pfahl and Wernli, 2012) and are the primary source of day-to-day weather variability in midlatitudes (Peixoto and Oort, 1992) and, consequently, have significant impacts on society. Understanding the strength of midlatitude cyclones requires understanding the mechanisms that drive their growth and decay, and much of what we know is based on theory for dry atmospheres (e.g., Charney, 1947; Eady, 1949; Lorenz, 1955; Salmon, 1980; O'Gorman and Schneider, 2008a; O'Gorman,

2010). However, Earth's atmosphere is not dry, and the presence of moisture also affects the dynamics of midlatitude weather systems.

One way moisture influences midlatitude cyclones is through latent heat release by condensation and precipitation. Condensation can enhance lower-level cyclonic potential vorticity (PV) anomalies and surface circulations and, in some cases, entirely replace the effect of low-level advection of PV across a meridional gradient (Kuo et al., 1991; Snyder and Lindzen, 1991; Davis

and Emanuel, 1991; Davis, 1992; Stoelinga, 1996; Ahmadi-Givi et al., 2004; Joos and Wernli, 2012; Tamarin and Kaspi, 2016). Additionally, latent heating can erode upper-level cylonic PV slightly downstream of the peak condensation and, in doing so, contribute to the westward phase tilt required for mutual Rossby wave amplification (Grams et al., 2011; Schemm et al., 2013). These effects matter: Emanuel et al. (1987) showed that baroclinic growth rates in a two-dimensional semigeostrophic model more than doubled when latent heating was included, and the growth rate can be even higher for diabatic Rossby vortices in

which latent heating is central to the dynamics (Kohl and O'Gorman, 2022). O'Gorman (2011) showed how a reduced effective static stability due to latent heating affects a number of aspects of the general circulation.

Although the importance of latent heating for midlatitude dynamics is well established, many atmospheric models neglect a different moisture effect that plays a role in dynamics and mass budgets: mass transports by hydrometeors (Trenberth, 1991; Trenberth et al., 1995; Qiu et al., 1993; Lackmann and Yablonsky, 2004; Wacker et al., 2006). Precipitation mass sinks reduce

hydrostatic surface pressure (Spengler et al., 2011), and this effect will be counteracted to some extent (depending on length scale) by convergent flow leading to vortex stretching. Like latent heating, falling hydrometeors act as sources and sinks of PV by moving mass across isentropic surfaces (Schubert et al., 2001; Lackmann and Yablonsky, 2004). However, while latent heat comprises about 2.5% of the total energy of the atmosphere (Peixoto and Oort, 1992), water vapor contributes only about 0.25% of the total mass of the atmosphere (Trenberth and Smith, 2005), and changes in atmospheric mass from water vapor

sources and sinks are ignored in many meteorological applications.

Nonetheless, some modeling studies have suggested that water vapor mass sources and sinks play a non-negligible role in the dynamics of heavily-precipitating weather systems. In simulations of an extratropical cyclone with a mesoscale model, Qiu et al. (1993) found that including precipitation mass sinks increased peak precipitation rates by about 10%. A similar effect was found for simulations of Hurricane Lili (2002): when mass sinks associated with precipitation were included, peak

10 meter wind speeds increased by between 5 and 15 knots, and the storm deepened by an additional 2-5 hPa (Lackmann and Yablonsky, 2004, LY04 hereafter). In the latter study, LY04 compared the production of PV by latent heating and mass sinks and found that, although local PV tendencies produced by latent heating were much larger than those produced by mass sinks, their vertical structures differed. Latent heating in Lili produced PV tendencies with vertically variable signs; mass sinks, on the other hand, produced single-signed PV increases. As a consequence, spatially-integrated PV tendencies from the

mass sink were non-negligible compared to those from latent heating, and LY04 argued that cancellation between opposite-signed PV tendencies produced by latent heating might allow the mass sink to play a comparatively non-negligible role in generating balanced circulations. Having both positive and negative PV tendencies from latent heating may decrease the effect of latent heating on storm strength either through transport, which can bring diabatically generated PV of one sign into a region of generation of the opposite sign (e.g., Schubert and Alworth, 1987; Büeler and Pfahl, 2017), or through inversion, which implies that winds are related to spatially smoothed PV anomalies (Hoskins et al., 1985).

Given that midlatitude baroclinic cyclones are significantly impacted by moisture, and in light of evidence that the precipitation mass sink may be important for the dynamics of some heavily precipitating systems, this paper investigates whether sources and sinks of water mass play a quantitatively significant role in weather variability and extremes in midlatitude storm tracks. Our work builds on Qiu et al. (1993) and LY04, but makes a number of novel contributions. First, we examine the importance of the mass sink both in climates similar to modern Earth and in much warmer and moister climates where mass sinks due to condensation and precipitation become significantly larger. Second, we focus primarily on the effect of mass sources and sinks on statistics of many storms in statistical equilibrium, not on the transient evolution of individual weather systems. Third, we develop theoretical results that provide insight into how PV sources and sinks, vertical transport, and inversion combine to influence the relative importance of latent heating versus mass sources and sinks. Finally, we consider the effects of including mass sources and sinks in tropical cyclone (TC) world simulations, providing a single-model comparison of the relative importance of mass sources and sinks for tropical and extratropical storms.

We proceed in three parts. First (Sections 2-3), we examine statistical measures of midlatitude variability and extremes in simulations across a broad range of climates using a widely-used idealized atmospheric general circulation model (Frierson et al., 2006, 2007; O'Gorman and Schneider, 2008b) that we modify to include sources and sinks of water mass. We find little difference between simulations with and without mass sources and sinks, even in climates much warmer than modern Earth, but do observe changes indicative of an increase in the strength of the strongest midlatitude cyclones when we exaggerate mass sources and sinks by a factor of 10. In Section 4, we connect simulation results with arguments based on isentropic potential vorticity, with a focus on explaining why effects from mass sources and sinks become apparent when mass sinks are exaggerated by a factor of 10 (and not, e.g., by a factor of 1 or 100). Finally, we attempt to reconcile our results with LY04 by examining the effect of mass sources and sinks on the intensity of TCs in TC world simulations in the same idealized GCM (Section 5). We find that adding mass sources and sinks can lead to transient deepening of some individual simulated TCs—similar to the results of LY04—but other TCs weaken, and systematic effects on statistics of large populations of TCs are relatively weak—similar to our results for midlatitude storm tracks.

## 2 Simulation Design

Our simulations use an idealized moist general circulation model based on the Geophysical Fluid Dynamics Laboratory (GFDL) spectral dynamical core. The model is documented in Frierson et al. (2006) and Frierson et al. (2007), with details largely following O'Gorman and Schneider (2008b); we briefly note some important aspects here.

The model uses two-stream gray radiative transfer with the insolation and longwave and shortwave absorber distributions described in O'Gorman and Schneider (2008b). We access a wide range of climates by varying a parameter $\alpha$ that multiplies the longwave optical thickness distribution from $\alpha = 1$, which gives a global-average surface temperature close to that of Earth today, to $\alpha = 6$, which gives a global-average surface temperature around 315 K. Surface fluxes of momentum and sensible and latent heat are calculated using standard drag laws. The lower boundary is a mixed layer of 1 meter depth whose temperature changes in response to radiation and surface turbulent fluxes. The dynamical core uses a spectral solver with $\nabla^8$ hyperdiffusion of vorticity, divergence, and temperature.

Condensation and precipitation are produced by a simplified Betts-Miller convection scheme (Frierson, 2007) and a large-scale condensation scheme. The convection scheme relaxes temperatures toward a moist adiabat and moisture toward 70% relative humidity on a two hour time scale, and ensures enthalpy conservation by modifying the time scale for specific humidity adjustment. The large-scale condensation scheme ensures that grid-scale relative humidity cannot exceed 100%. Unlike O'Gorman and Schneider (2008b), we include reevaporation of falling condensate, which is assumed to bring subsaturated grid cells to saturation before falling through them. The convection and large-scale condensation schemes both assume a pseudoadiabatic limit in which water is removed from the atmosphere as soon at it condenses.

Our main modification to the model is the inclusion of mass sources and sinks associated with evaporation and condensate fallout, respectively. The changes are described in detail in Appendix A, but we summarize them here. Briefly, we add terms to equations for pressure tendencies at the surface and at interfaces between levels to represent the effects of adding and removing mass from the overlying column (Appendix A1). Our implementation scales pressure tendencies from mass sources and sinks by a parameter $\gamma$. Setting $\gamma = 0$ corresponds to neglecting pressure tendencies from mass sources and sinks, setting $\gamma = 1$ corresponds to including the correct pressure tendencies from mass sources and sinks, and setting $\gamma > 1$ exaggerates pressure tendencies from mass sources and sinks. Note that $\gamma$ modifies pressure tendencies from mass sources and sinks, but does not alter humidity tendencies themselves. This means that setting $\gamma = 0$ implies that sinks of moist mass are exactly offset by sources of dry mass, and that setting $\gamma > 1$ implies that sinks of moist mass are accompanied by additional sinks of dry mass. Numerical errors in our implementation of the mass sinks lead to a slow downward drift in total mass, and so our simulations use a dry mass fixer that ensures conservation of a fixed dry mass inventory of $10^5$ Pa (Appendix A3). For simplicity, we refer to the dry mass inventory as "dry mass" and the additional atmospheric mass due to water vapor as "moist mass" for the remainder of the paper. We do not include energy or moisture fixers in our simulations, as neither are required to achieve statistical steady states. Finally, we include minor changes to the calculation of the reference moist adiabat used by the convection scheme to improve consistency with the model dynamical core (Appendix A4), but these are unrelated to the addition of mass sources and sinks.

The form of the mass sink term used in our simulations relies on the pseudoadiabatic assumption, which is consistent with the model physics but nonetheless approximate. In the pseudoadiabatic limit, water mass disappears immediately upon condensation, and and the resulting mass loss has an immediate impact on the hydrostatic pressure field at all levels below the level where condensation occurs. An additional consequence of our use of a pseudoadiabatic model is that our simulations contain no representation of the effects of condensate loading on the hydrostatic pressure field. In reality, mass sinks are

associated not with condensation itself, but with subsequent hydrometeor fallout (e.g., Ooyama, 2001), and condensate loading from cloud water and sedimenting hydrometeors contributes to the total hydrostatic pressure. As shown in Appendix B, the pseudoadiabatic mass sink term can be derived from a mass sink term that includes condensate loading and explicit hydrometeor fallout by assuming that time scales for precipitation formation and fallout are infinitely fast. Because our study focuses primarily on the dynamics of midlatitude eddies and tropical cyclones (large-scale weather systems that evolve over time scales of days), we think that treating precipitation formation and fallout as infinitely fast is a reasonable simplifying assumption. Our use of the pseudoadiabatic limit is likely to affect some aspects of our simulations (for example, the tropics may be nearly neutral to pseudoadiabatic ascent, rather than to reversible adiabatic ascent as found in Xu and Emanuel (1989)), but we do not think that the use of a full cloud microphysics scheme or prognostic equations for condensed water species are essential for our study, which focuses on the existence or absence of the mass sink itself. Nevertheless, we would welcome follow-up studies that examine whether a more sophisticated treatment of clouds and precipitation alters any of our conclusions.

We use T170 horizontal resolution (nominal grid spacing of about 80 km at the equator) and 30 unevenly-spaced sigma levels for all simulations, and access a wide range of climates by running sets of simulations with $\alpha = (1, 2, 3, 4, 5, 6)$. All of our simulations are spun up from an isothermal rest state for 700 days, and statistics are calculated over the following 300 days based on instantaneous snapshots taken every 6 hours. We refer to $\alpha = 1$ simulations, which produce global-mean surface air temperatures close to modern values, as "control" simulations, and simulations with $\alpha > 1$ as "warm" simulations. At each $\alpha$, we run simulations with three difference treatments of mass sources and sinks. One treatment neglects mass sources and sinks ($\gamma = 0$, "no MS"), one includes mass sources and sinks ($\gamma = 1$, "MS"), and one exaggerates mass sources and sinks by a factor of 10 ($\gamma = 10$, "10x MS"). Total atmospheric mass remains equal to the dry mass inventory in all no MS simulations, increases by as much as 3% in warm MS simulations, and increases by between about 3% and 25% (depending on temperature) in 10x MS simulations (Figure 1). The mass increases in the MS and 10x MS simulations because of the source of mass from evaporation, which comes into balance with the precipitation mass sink at statistical steady state, and this evaporation mass source is larger again in the 10x-MS simulations.

## 3   Impact of the Mass Sink on Midlatitude Storm Tracks

Our simulations, in both control and warm climates and for all treatments of the mass sink, produce midlatitude storm tracks with pronounced eddy activity and a relatively quiescent tropics (Figure 2). The character of the eddy field, as indicated by near-surface wind speeds, changes significantly between the control climate ($\alpha = 1$, Figure 2a-c) and the warmest climate ($\alpha = 6$, Figure 2d-f). In the control climate, strong near-surface winds are widespread across midlatitude baroclinic zones. In the warmest climate, eddies appear less vigorous: near-surface winds are strong only in localized eddies and relatively weak elsewhere. Generally-weaker near-surface winds in warmer climates are a symptom of reduced midlatitude eddy kinetic energy, which decreases in the warmest climates because weaker meridional temperature gradients and higher midlatitude dry static stability reduce mean available potential energy (O'Gorman and Schneider, 2008a). Regions of strong near-surface winds may appear more localized in warmer climates because of changes to the dominant mode of midlatitude instability (O'Gorman

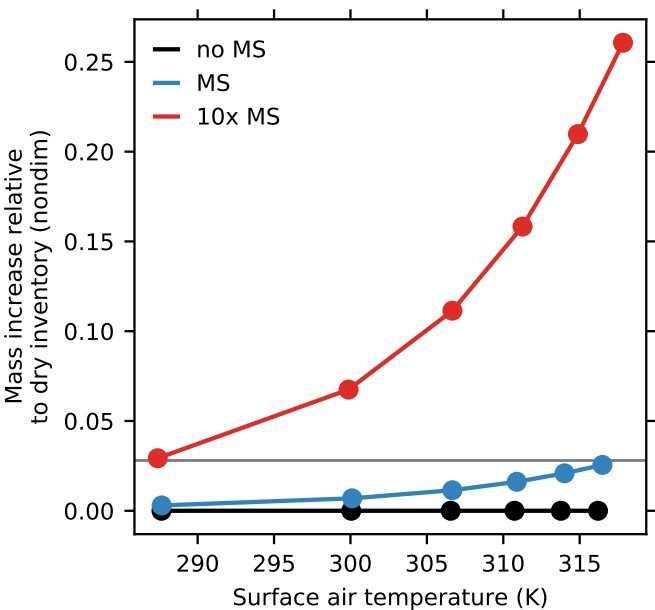

**Figure 1.** Ratio of moist atmospheric mass to dry atmospheric mass, plotted as a function of global-mean air temperature in the lowest model level. The horizontal gray line is a visual aid for comparing moist mass in the warmest MS simulation and control 10x-MS simulation.

et al., 2018; Kohl and O'Gorman, 2022). In control simulations, precipitation is strongest in frontal zones, where precipitation rates of 1-3 mm hour$^{-1}$ are typical (Figure 3a-c). In the warmest simulations, precipitation rates are highest near the center of cyclones and often reach values above 10 mm hour$^{-1}$ (Figure 3d-f).

We focus on statistics calculated over midlatitude baroclinic zones, defined following O'Gorman and Schneider (2008a) as regions within 15 degrees latitude of the maximum of the vertically-integrated eddy potential temperature flux $\overline{v'\theta'}\cos\phi$, where $v$ is meridional velocity, $\theta$ is potential temperature, $\phi$ is latitude, $(\cdot)'$ is an eddy quantity defined relative to the surface-pressure-weighted zonal and time mean $\overline{(\cdot)}$ along $\sigma$ surfaces, and the vertical integral is taken from the surface to the lowest level where the zonal- and time-mean temperature lapse rate drops below 2 K km$^{-1}$. The baroclinic zones track regions of strong eddy activity and shift poleward in warmer climates (Figure 2). We compute 95% confidence intervals for all statistics by resampling raw model output in time using block bootstrapping with a block size of 10 days. We report confidence intervals based on 200 block bootstrap realizations, each with 300 days of data. The confidence intervals are robust to changes in the number of bootstrap realizations; using 100 or 50 bootstrap realizations modifies them only slightly.

Figure 4 shows vertically-averaged eddy kinetic energy (EKE), near-surface EKE, and surface pressure variance averaged over the midlatitude baroclinic zones. Vertically-averaged EKE is averaged over all model levels, and near-surface EKE is defined as EKE in the lowest model level. Vertically-integrated and near-surface EKE both decrease with warming (O'Gorman and Schneider, 2008a), and values from simulations with different treatments of the mass sink are statistically indistinguishable. Like EKE, surface pressure variance decreases with warming and is statistically indistinguishable in no-MS and MS

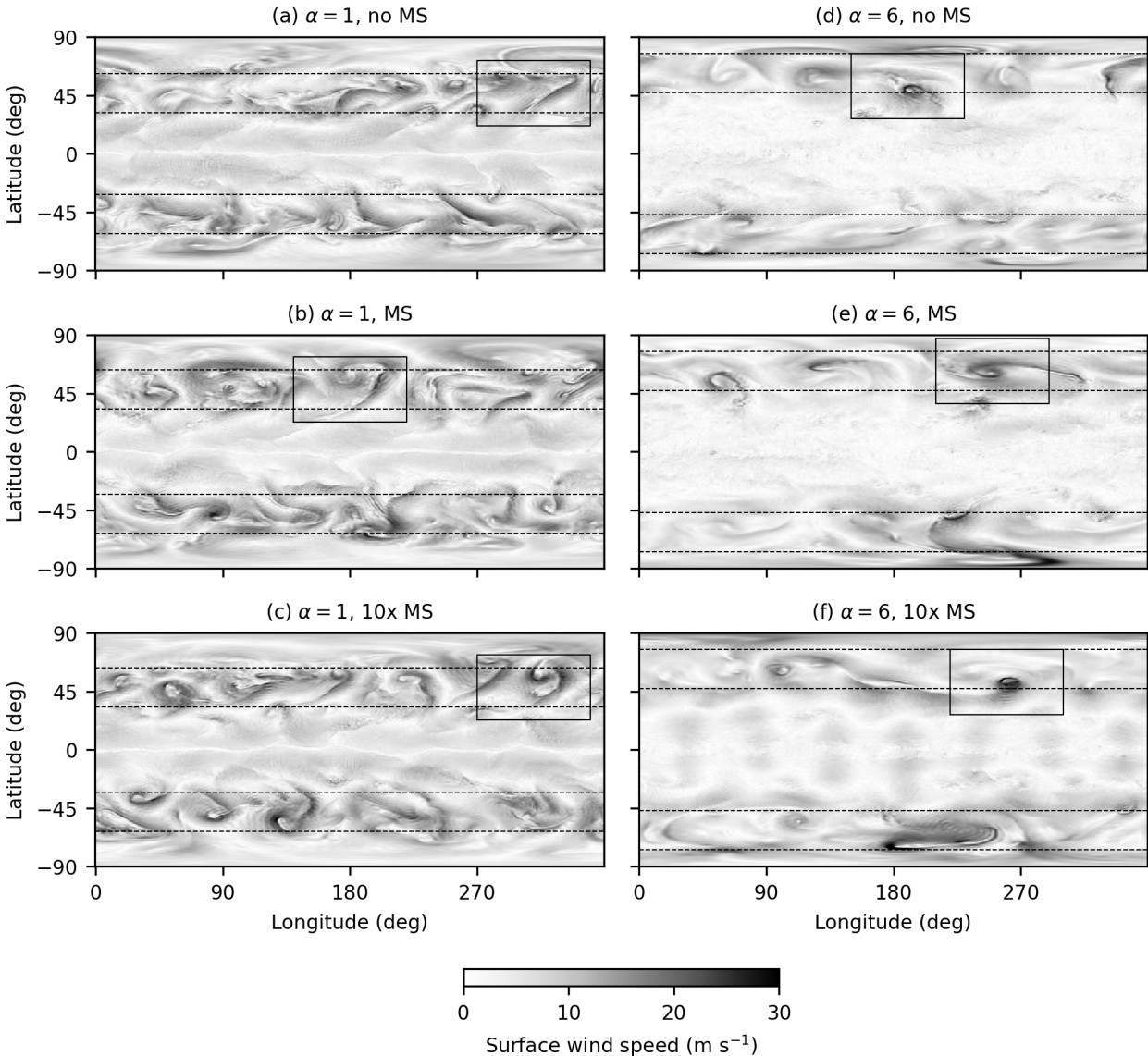

**Figure 2.** Wind speed in the lowest model level from the last saved snapshot of simulations in the control climate with $\alpha = 1$ (a,b,c) and the warmest climate with $\alpha = 6$ (d,e,f) with no mass sinks (a,d), realistic mass sinks (b,e), and mass sinks exaggerated by a factor of 10 (c,f). Midlatitude baroclinic zones defined based on the meridional eddy potential temperature flux are delimited by dashed lines, and rectangles delimit regions for which precipitation rates are shown in Figure 3.

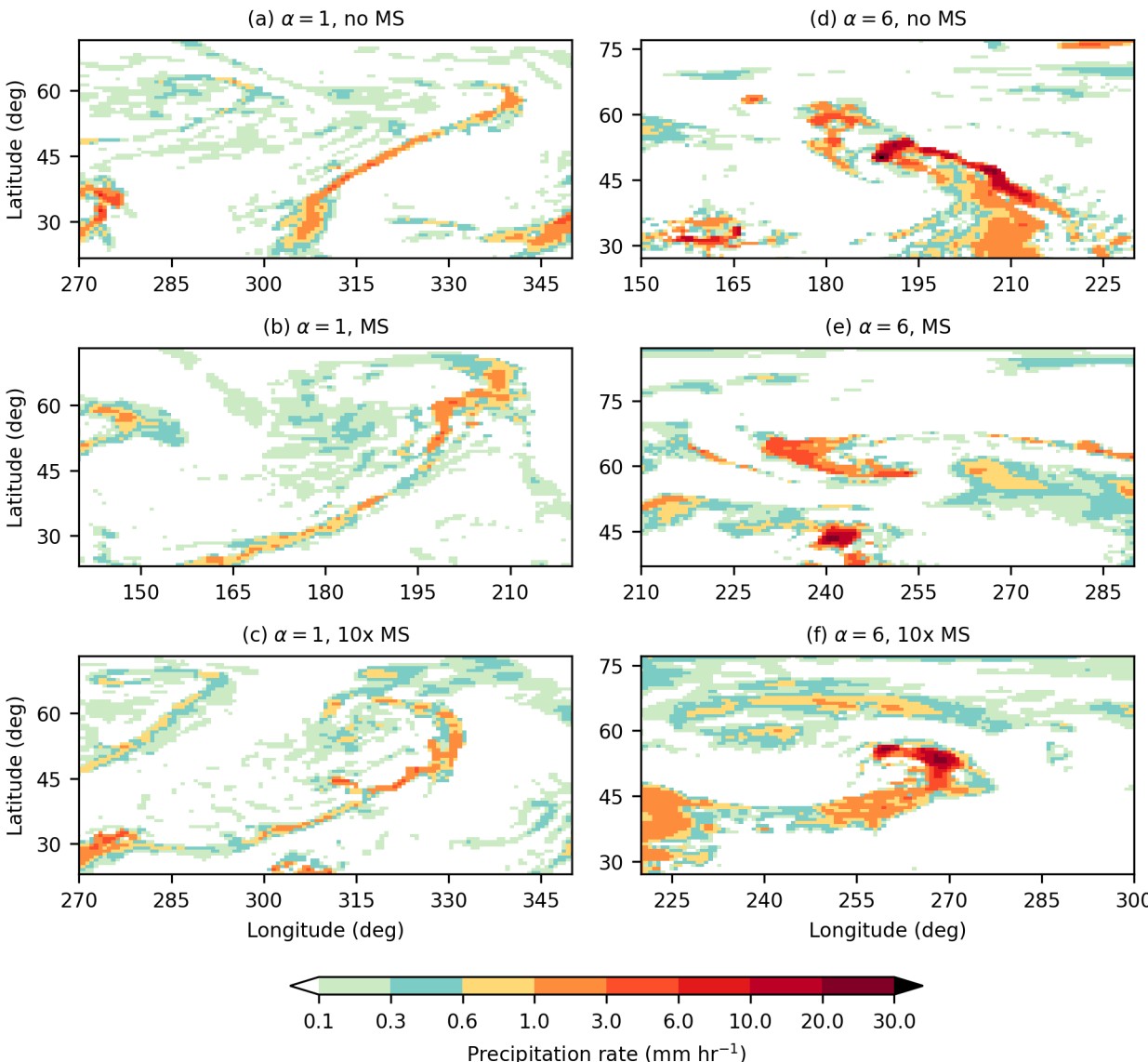

**Figure 3.** Surface precipitation rates from the last saved snapshot of simulations in the control climate with $\alpha = 1$ (a,b,c) and the warmest climate with $\alpha = 6$ (d,e,f) with no mass sinks (a,d), realistic mass sinks (b,e), and mass sinks exaggerated by a factor of 10 (c,f). Regions shown are outlined in rectangles in Figure 2.

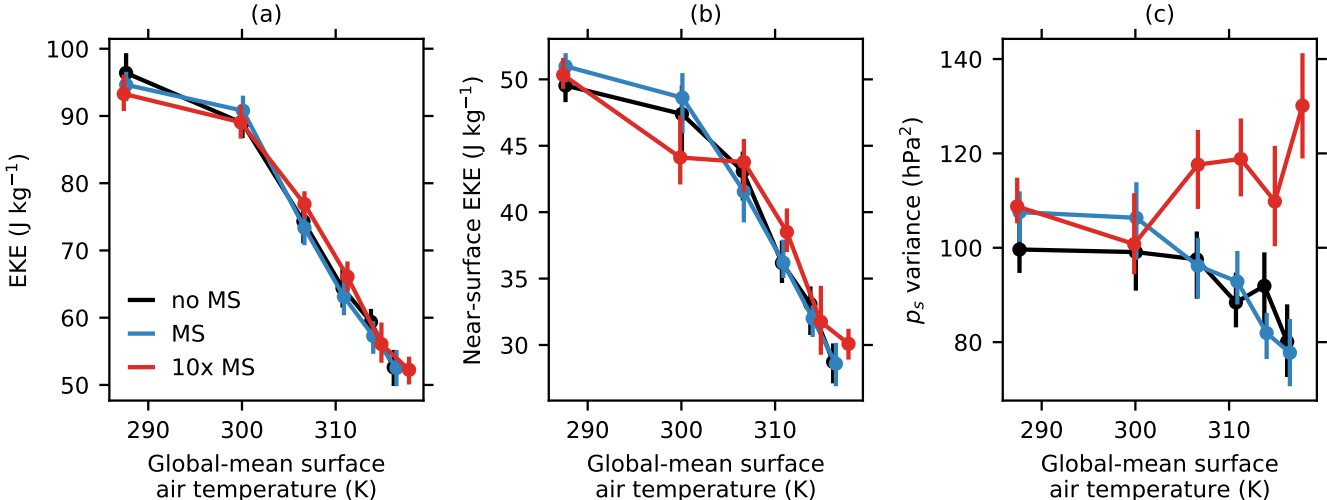

**Figure 4.** Vertically-integrated EKE (a), near-surface EKE (b), and surface pressure variance (c) averaged over midlatitude baroclinic zones and plotted as a function of global-mean surface air temperature. Vertical lines indicate 95% confidence intervals computing using block bootstrap resampling as described in the text. Horizontal lines indicating 95% confidence intervals for global-mean surface temperature are also included but are too small to be visible.

simulations. In 10x-MS simulations, however, surface pressure variance increases with warming, and is substantially higher than in no MS and MS simulations at $\alpha = 3$ and above (global-mean surface temperatures warmer than $\sim 305$ K). No-MS and 10x-MS simulations can differ in in surface pressure variance even though they do not differ in EKE because large mass sinks produce variations in surface pressure with an unbalanced component that is not associated with large-scale fluid flow.

Figure 5 shows extreme percentiles of near-surface wind speeds, surface pressure anomalies, and (instantaneous) precipitation in midlatitude baroclinic zones. In all cases, percentiles are calculated as a function of latitude (aggregating over longitude and time), then combined in area-weighted latitudinal averages over the baroclinic zones. Precipitation percentiles are calculated for all times, not just times when rain occurs. 99th percentile near-surface wind speeds, defined as wind speeds in the lowest model level, decrease with warming and are statistically indistinguishable in no-MS and MS simulation, but are systematically higher in 10x-MS simulations. Extreme surface lows, calculated as the 1st percentile of surface pressure anomalies relative to zonal- and time-averages, are likewise statistically indistinguishable in no-MS and MS simulations, but systematically deeper in 10x-MS simulations in warm climates. The behavior of 99.9th percentile rain rates mirror that of extreme wind speeds and surface lows: no-MS and MS simulations are again statistically-indistinguishable, but 10x-MS simulations produce systematically stronger precipitation in warm climates. 99.9th percentile precipitation rates increase with warming, as expected from simple arguments about the influence of warming on extreme precipitation (O'Gorman and Schneider, 2009; Muller et al., 2011). Lower precipitation percentiles (e.g., 99th percentile precipitation rates) decrease with warming in the warmest climates (not shown), suggesting that increases in 99th percentile precipitation rates may be limited to some extent

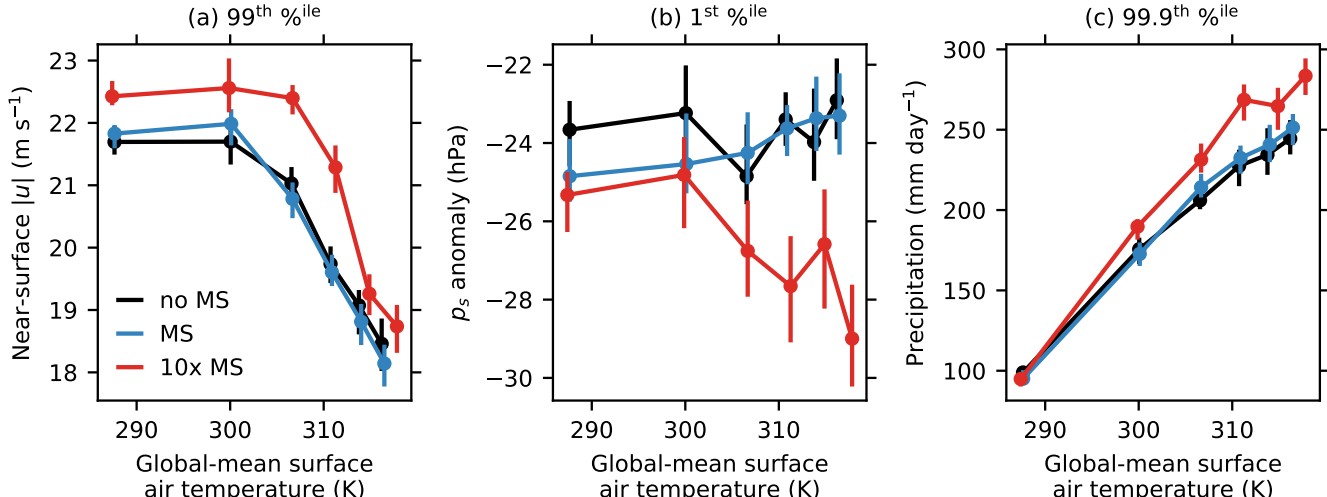

**Figure 5.** Extreme near-surface wind speeds (a), surface pressure anomalies (b), and precipitation (c) calculated over midlatitude baroclinic zones and plotted as a function of global-mean surface air temperature. 99th and 99.9th percentiles in panels a and c highlight particularly strong near-surface wind and precipitation; 1st percentiles shown in panel (b) highlight particularly strong surface lows. Vertical lines indicate 95% confidence intervals computed using block bootstrap resampling as described in the text. Horizontal lines indicating 95% confidence intervals for global-mean surface temperature are also included but are too small to be visible.

by energetic constraints on mean precipitation (O'Gorman and Schneider, 2008b; Pendergrass and Hartmann, 2014a, b) or by decreases in mean midlatitude eddy kinetic energy (O'Gorman and Schneider, 2008a).

To summarize: statistics calculated over midlatitude baroclinic zones are consistently statistically indistinguishable in no-MS and MS simulations, but noticeably different between no-MS and 10x-MS simulations, particularly in warm climates. Differences between no-MS and 10x-MS simulations suggest that extreme disturbances as well as the unbalanced component

of surface pressure anomalies become more intense with a tenfold-exaggerated mass sink: midlatitude surface pressure variance increases, extreme near-surface winds become faster, extreme surface lows become deeper, and extreme precipitation strengthens. In contrast to other statistics, mean midlatitude EKE does not show a statistically-significant difference between no-MS and 10x-MS simulations, possibly because of constraints imposed by the mean available potential energy that apply to the mean EKE of baroclinic zones (O'Gorman and Schneider, 2008a) but not to individual weather systems or to unbalanced

pressure variations. The overall weight of evidence points to minor differences in storm track variability and extremes between no-MS and MS simulations, but noticeable increases in the strength of extreme disturbances and surface pressure variability in 10x-MS simulations.

Our results, which shows no detectable differences between no-MS and MS simulations even in climates much warmer than the modern, are seemingly inconsistent with the results of LY04, which document a clear increase in TC intensity in a

210 simulation of Hurricane Lilli with mass sinks included. This inconsistency, plus the fact that we do find differences between no-MS and 10x-MS simulations, prompts us to better understand the underlying theory and ask a question: is there reason

to expect the dynamical effects of unexaggerated mass sinks to be dominated by other processes, but to become at least marginally detectable when exaggerated by a factor of 10? In the next section, we use some simple theoretical analysis of isentropic potential vorticity equations to argue that there is.

## 4  Isentropic Potential Vorticity Theory

In this section, we analyze isentropic potential vorticity (IPV) equations to probe why flows in midlatitude baroclinic zones are insensitive to the inclusion of mass sources and sinks but affected by tenfold-exaggerated mass sources and sinks. Following LY04, our approach is to compare the effects of mass sinks with the effects of latent heating, the latter being a convenient benchmark that generally is considered important to include in weather and climate models. We first show that the magnitude of IPV sources produced by latent heating is larger than those produced by mass sinks by a factor of about 200 (Section 4.1). This factor is influenced by thermodynamic constants and tropospheric stratification, but is not directly climate-dependent because latent heating and mass sinks increase in tandem with warming. We then analyze the evolution of IPV anomalies produced by sustained latent heating with a first-baroclinic-mode vertical structure, which produces an IPV source in the lower troposphere and an IPV sink in the upper troposphere. We show that the combined effects of latent heating and vertical advection produce, at long times, a positive IPV anomaly throughout most of the depth of the atmosphere; and that mass sinks increase the positive IPV anomaly only by about 1 part in 200 (Section 4.2). At short times, cancellation between opposite-signed IPV anomalies produced by latent heating can limit the strength of induced flow because the inversion is non-local, but we show that flows induced by latent heating remain stronger than flows induced by mass sinks by a factor of 100 or more except for disturbances with horizontal length scales much larger than a deformation radius (Section 4.3).

For consistency with our simulations, our IPV analysis assumes a pseudoadiabatic limit where condensation falls out of the atmosphere immediately upon formation, which influences the form in which mass sinks appear as IPV source terms. As discussed previously, we view our use of a pseudoadiabatic limit as a reasonable choice given the focus of our work. However, we include in Appendix D a derivation of an IPV equation that does not assume pseudoadiabatic fallout (in which case the IPV source from mass sinks involves the divergence of the hydrometeor sedimentation mass flux), and show that the pseudoadiabatic IPV equation used in this section can be recovered by assuming that precipitation formation and fallout are infinitely fast.

### 4.1  IPV Tendencies from Latent Heating and Mass Sinks

We begin with frictionless momentum and continuity equations in potential temperature coordinates on an f-plane (Holton and Hakim, 2012):

$$\frac{\partial \mathbf{v}}{\partial t} + \nabla_\theta \left( \frac{\mathbf{v} \cdot \mathbf{v}}{2} + M \right) + (\zeta_\theta + f)\, \mathbf{k} \times \mathbf{v} = -\dot{\theta} \frac{\partial \mathbf{v}}{\partial \theta} \tag{1}$$

$$\frac{\partial \sigma}{\partial t} + \nabla_\theta \cdot (\sigma \mathbf{v}) + \frac{\partial}{\partial \theta} \left( \sigma \dot{\theta} \right) = \sigma \dot{q}_v. \tag{2}$$

$\theta$ is potential temperature, $\mathbf{v}$ is the horizontal velocity, $\nabla_\theta$ is the horizontal gradient on potential temperature surfaces, $\zeta_\theta = \mathbf{k} \cdot \nabla_\theta \times \mathbf{v}$ is the relative vorticity, and $f$ is the Coriolis parameter. $M = c_p T + gz$ is the Montgomery streamfunction, with $c_p$ the specific heat capacity of dry air, $T$ the absolute temperature, $g$ the acceleration from gravity, and $z$ geometric height. $\sigma = -g^{-1}\partial p/\partial\theta$ is the isentropic density, with $p$ the pressure. $\dot\theta$ is a diabatic potential temperature tendency, used here to represent the effect of latent heat release, and $\dot q_v$ is the specific humidity tendency from condensation and precipitation fallout. We consider a pseudoadiabatic limit where all condensation is immediately removed from the atmosphere, consistent with model simulations described earlier, so $\dot q_v$ corresponds directly to a mass sink. A derivation of the continuity equation including mass sources and sinks is provided in Appendix C; the version used here approximates $1 - q_v \approx 1$.

Following Holton and Hakim (2012), we take $\mathbf{k} \cdot \nabla_\theta \times$ of the momentum equation to obtain

$$\left(\frac{\partial}{\partial t} + \mathbf{v} \cdot \nabla_\theta\right)(\zeta_\theta + f) + (\zeta_\theta + f)\nabla_\theta \cdot \mathbf{v} = -\mathbf{k} \cdot \nabla_\theta \times \left(\dot\theta\frac{\partial\mathbf{v}}{\partial\theta}\right), \tag{3}$$

then expand the right-hand-side derivative to rewrite the result as

$$\frac{\mathrm{D}}{\mathrm{D}t}(\zeta_\theta + f) + (\zeta_\theta + f)\nabla_\theta \cdot \mathbf{v} = \mathbf{k} \cdot \left(\frac{\partial\mathbf{v}}{\partial\theta} \times \nabla_\theta\dot\theta\right), \tag{4}$$

where

$$\frac{\mathrm{D}}{\mathrm{D}t} = \frac{\partial}{\partial t} + \mathbf{v} \cdot \nabla_\theta + \dot\theta\frac{\partial}{\partial\theta} \tag{5}$$

is the material derivative expressed in potential temperature coordinates. Using the continuity equation to eliminate the stretching term gives

$$\frac{\mathrm{D}P}{\mathrm{D}t} = P\left(\frac{\partial\dot\theta}{\partial\theta} - \dot q_v\right) + \sigma^{-1}\mathbf{k} \cdot \left(\frac{\partial\mathbf{v}}{\partial\theta} \times \nabla_\theta\dot\theta\right), \tag{6}$$

where

$$P = \frac{\zeta_\theta + f}{\sigma} \tag{7}$$

is the IPV. The source terms involving $\dot\theta$ and $\dot q_v$ are similar to source terms for Ertel potential vorticity (EPV) derived by Schubert et al. (2001), and the IPV evolution equation is identical to that given in Hoskins (1991) except that we assume the flow is frictionless and include an IPV source term for mass sources and sinks.

Next, we assume we are at a horizontal maximum in latent heating such that the final term in Equation 6 vanishes. We further assume that the latent heating profile is given by

$$\dot\theta = \frac{\theta_t - \theta_s}{T}\sin\left(\pi\frac{\theta - \theta_s}{\theta_t - \theta_s}\right), \tag{8}$$

where $\theta_s$ and $\theta_t$ are the potential temperature at the surface and the top of the region of latent heating and $T$ is a time scale. In the pseudoadiabatic limit, the precipitation mass sink is related to the latent heating rate by

$$\dot q_v = -\frac{\Pi}{L_v}\dot\theta, \tag{9}$$

where $L_v$ is the latent heat of vaporization for water and

$$\Pi = \frac{c_p T}{\theta} = c_p \left( \frac{p}{p_s} \right)^{R/c_p} \tag{10}$$

is the Exner function, with $p_s$ the surface pressure and $R$ the gas constant for dry air. Substituting these expressions for $\dot{\theta}$ and $\dot{q}_v$ into Equation 6 and non-dimensionalizing by setting

$$t \to T\tilde{t}$$

$$\theta - \theta_s \to (\theta_t - \theta_s)\tilde{\theta}$$

$$P \to \frac{f}{\sigma_0}\tilde{P}$$

$$\Pi \to c_p \tilde{\Pi},$$

where $\sigma_0$ is a reference isentropic density, gives

$$\frac{\mathrm{D}\tilde{P}}{\mathrm{D}\tilde{t}} = \tilde{P}\left[ \pi \cos\left(\pi\tilde{\theta}\right) + \frac{c_p(\theta_t - \theta_s)}{L_v}\tilde{\Pi}\sin\left(\pi\tilde{\theta}\right) \right]. \tag{11}$$

The ratio of the first and second bracketed terms in Equation 11, which gives the ratio of the IPV source from latent heating to the IPV source from mass sinks, scales like

$$A = \frac{\pi L_v}{c_p(\theta_t - \theta_s)} \approx 200, \tag{12}$$

using 40 K as estimate of $\theta_t - \theta_s$ appropriate for a latent heating profile that extends through the entire depth of the midlatitude troposphere. The expression for and approximate value of $A$—a simple measure of the relative magnitude of PV tendencies

from latent heating and mass sinks—are the main results of this section. The value of $A$ suggests that IPV anomalies produced by latent heating are likely to be about 200 times larger than IPV anomalies produced by mass sinks. Additionally, the fact that $A$ depends only thermodynamic constants and the potential temperature range within the region of latent heating suggests that its value is unlikely to vary strongly with climate. Both features are consistent with there being no detectable differences in cyclone statistics in our simulations without a mass sink and with the mass sink included. Our estimated value of $A \approx 200$ is also

consistent with lower-tropospheric (Ertel) potential vorticity sources reported by LY04 for Hurricane Lili: in their simulation, lower-tropospheric PV sources from latent heating peak at around 25 PV units (PVU) per hour, while lower-tropospheric sources from mass sinks peak at around 3-4 PVU day$^{-1}$. PV sources from mass sinks reported by LY04 are somewhat larger near the melting line, likely because of microphysical effects not considered in our theory, but remain about 70-80 times smaller than PV sources from latent heating.

Finally, we note that the value of $A$ provides a plausible explanation for simulation results showing quantitatively detectable (albeit still minor) differences in midlatitude cyclone statistics between simulations without a mass sink and simulations with a tenfold-exaggerated mass sink. With a tenfold exaggeration of mass sinks relative to latent heating, our estimate of the ratio of IPV tendencies from latent heating and mass sinks is reduced to $A/10 \approx 20$. This value, although still significantly larger than

one, is small enough to suggest that the dynamical effects of mass sinks may not remain completely insignificant compared to the effects of latent heating: a 5% (1/20) increase to the diabatic PV source is not a first-order effect, but not obviously negligible either.

## 4.2 Evolution of IPV anomalies

Our scale analysis in the previous section suggests that IPV tendencies from precipitation mass sinks are about a factor of 200 smaller than IPV tendencies produced by associated latent heating. However, IPV tendencies produced by latent heating and mass sinks have very different characteristic vertical structures: first-baroclinic latent heating produces an IPV source in the lower troposphere and an IPV sink in the upper troposphere, whereas the associated mass sink produces an IPV source everywhere. In terms of the effect on the induced winds, having different signs of IPV sources from latent heating may lead to weaker winds because the inversion from IPV to winds is spatially non-local and because vertical transport of IPV may bring diabatically-generated IPV into a region where the diabatic generation is of a different sign.This effect is highlighted by LY04, who emphasize the role of spatial cancellation in limiting volume-integrated potential vorticity tendencies from latent heating. In this and the following section, we examine the ramifications of having regions with different-signed IPV sources from latent heating for the relative importance of latent heating versus mass sinks. We start by examining the temporal evolution of potential vorticity anomalies produced by the idealized latent heating and mass sink profiles introduced in the previous section.

Beginning with Equation 11 and neglecting horizontal advection gives

$$\frac{\partial \tilde{P}}{\partial \tilde{t}} = -\sin\left(\pi\tilde{\theta}\right)\frac{\partial \tilde{P}}{\partial \tilde{\theta}} + \tilde{P}\pi\cos\left(\pi\tilde{\theta}\right) + \tilde{P}\frac{c_p\left(\theta_t - \theta_s\right)}{L_v}\tilde{\Pi}\sin\left(\pi\tilde{\theta}\right). \tag{13}$$

We retain vertical advection (first term on the right-hand side) because vertical advection of anomalous potential vorticity plays an important role in setting potential vorticity profiles in mature cyclones (Schubert and Alworth, 1987; Büeler and Pfahl, 2017). The non-dimensional Exner function $\tilde{\Pi} = (p/p_s)^{R/c_p}$ should in principle vary in time as a circulation develops and $p(\theta)$ changes, but because $\tilde{\Pi}$ only varies from about 1 to 0.5 as $p$ varies from $p_s$ to $p_s/10$, we approximate $\tilde{\Pi}$ as fixed in time. We still need to specify its vertical structure, however, and for simplicity we assume that pressure varies linearly from $p_s$ at $\theta = \theta_s$ to $(1 - f_p)p_s$ at $\theta = \theta_t$, giving $p = p_s\left(1 - f_p\tilde{\theta}\right)$ and

$$\tilde{\Pi} = \left(1 - f_p\tilde{\theta}\right)^{R/c_p}. \tag{14}$$

We integrate Equation 13 starting from $\tilde{P} = 1$ at $\tilde{t} = 0$ (a quiescent initial condition with zero relative vorticity) and ending at $\tilde{t} = 2$. We set $\theta_t - \theta_s = 40$ K, corresponding to $A \approx 200$, and $f_p = 0.7$, corresponding to latent heating that extends from 1000 hPa to 300 hPa. The solution, which we plot as an anomaly relative to the initial condition and denote $\tilde{P}'$, initially shows positive lower-tropospheric and negative upper-tropospheric IPV anomalies of equal depths (Figure 6a). The opposite-signed anomalies are produced by latent heating (second term on the right-hand side of Equation 13). However, the depth of the negative upper-tropospheric anomaly rapidly decreases, and negative IPV anomalies grow much less quickly than positive IPV anomalies. Vertical advection (first term on the right-hand side of Equation 13), which transports positive IPV anomalies upward into regions with an IPV sink from latent heating, is responsible for decreasing the depth of the negative IPV anomaly.

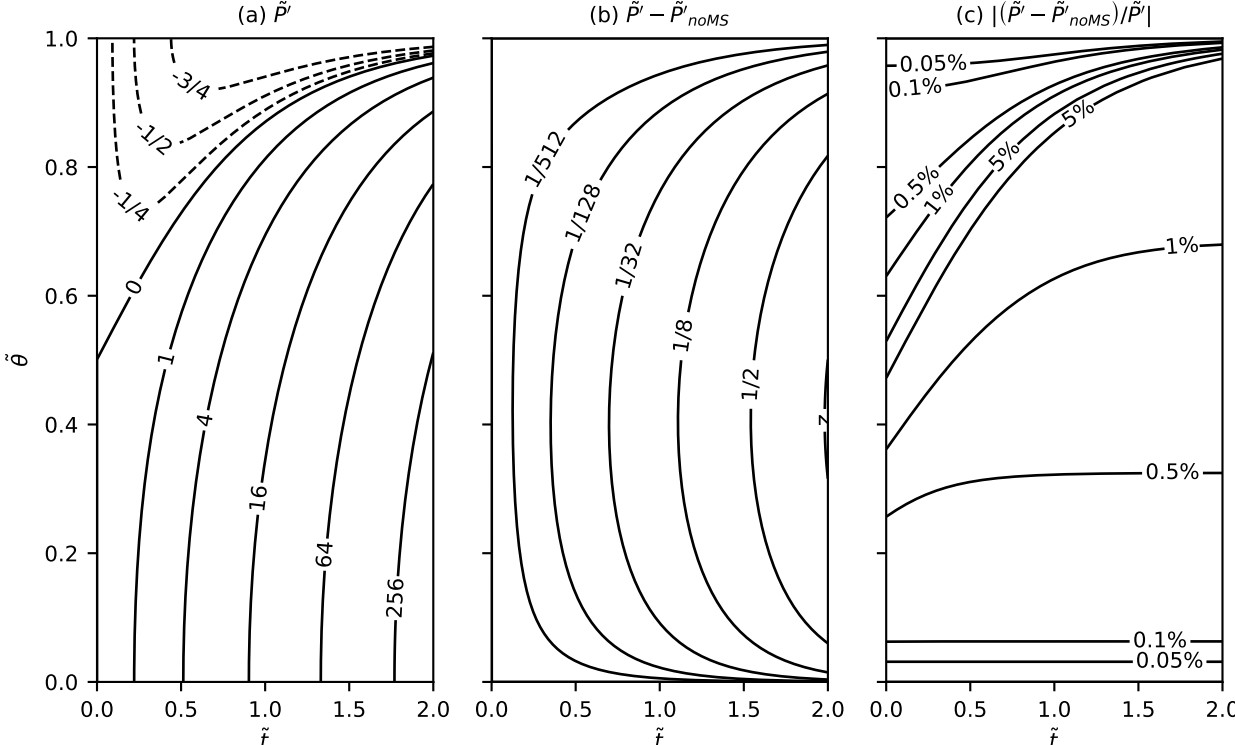

**Figure 6.** Evolution of IPV in a precipitating column and the contribution from the precipitation mass sink. Panel (a) shows the IPV anomaly $\tilde{P}'$ that develops in response to latent heating, mass sinks, and vertical advection. Panel (b) shows the difference between $\tilde{P}'$ and the IPV anomaly $\tilde{P}'_{noMS}$ that develops with only latent heating and vertical advection included. This difference quantifies the influence of the mass sink on $\tilde{P}'_{both}$. Panel (c) shows the magnitude of the relative difference between IPV anomalies that develop with and without the mass sink.

The factor of $\tilde{P}$ multiplying IPV sources and sinks (second and third terms right-hand side terms in Equation 13) is responsible for limiting the growth of negative IPV anomalies, because it causes $\tilde{P}$ to decay exponentially toward 0 in response to a sustained IPV sink, whereas positive IPV anomalies can grow without bound.

We quantify the influence of mass sinks on $\tilde{P}'$ by integrating Equation 13 a second time with the mass sink IPV source (third right-hand side term) removed. Subtracting the resulting IPV anomaly, which we denote $\tilde{P}'_{noMS}$, from $\tilde{P}'$ shows that the mass sink contributes a positive IPV anomaly throughout the entire depth of the atmosphere (Figure 6b). However, the contribution of the mass sink is small compared to the total IPV anomaly. Plotting $\left| \left( \tilde{P}' - \tilde{P}'_{noMS} \right) / \tilde{P}' \right|$ (i.e., the magnitude of the relative difference between anomalies produced with and without mass sinks) shows that mass sink contributes only about a fraction $0.5\% \approx 1/A$ of the total positive IPV anomaly. In the lower troposphere ($\tilde{\theta} < 0.5$), a positive IPV tendency from upward transport and an IPV source from latent heating both contribute to the positive IPV anomaly and are both significantly larger than the IPV source from mass sinks. In the upper troposphere ($\tilde{\theta} > 0.5$), the combined positive tendency from upward

transport plus negative tendency from latent heating eventually becomes positive and significantly larger than the IPV source from mass sinks. There is always a thin layer (initially near $\tilde{\theta} = 0.5$, and higher at later times) where the net IPV anomaly produced by compensating sinks from latent heating and sources from upward transport is small, and the relative difference between anomalies produced with and without mass sinks is fairly large (region between the 5% contours in Figure 6c). At long times ($\tilde{t} \gtrsim 1$), however, a positive IPV anomaly extends throughout most of the depth of the atmosphere, and the fractional contribution of mass sinks to the overall PV anomaly is small everywhere except in a thin layer near the top of the domain where the anomaly changes sign. So, despite the *initial* positive and negative IPV anomalies produced by latent heating being of opposite sign in the lower and upper troposphere, the long-term evolution of the IPV field is toward a vertically-extensive positive IPV anomaly produced almost entirely by latent heating plus vertical advection, with only a very small fractional contribution from mass sinks.

### 4.3 Spatial Cancellation during Inversion

At short times ($\tilde{t} << 1$), however, IPV anomalies produced by latent heating do exhibit positive and negative regions of similar magnitude. Because of the nonlocal relationship between IPV anomalies and induced circulations, this dipole structure has the potential to limit the strength of circulations induced by latent heating. In this section, we use linearized IPV inversions to examine whether the relative strength of *circulations* induced by latent heating and mass sinks scales with the IPV tendency ratio $A$, or whether spatial cancellation of IPV anomalies during inversion significantly weakens circulations induced by latent heating relative to those induced by mass sinks.

Substituting $\tilde{P} = \tilde{P}' + 1$ into Equation 13, assuming that $\tilde{P}' << 1$ (valid at short times), and dropping right-hand-side terms that involve $\tilde{P}'$ gives a short-time approximation to Equation 13:

$$\frac{\partial \tilde{P}'}{\partial \tilde{t}} = \pi \cos\left(\pi \tilde{\theta}\right) + \frac{c_p \left(\theta_t - \theta_s\right)}{L_v} \tilde{\Pi} \sin\left(\pi \tilde{\theta}\right). \tag{15}$$

Assuming that the strength of latent heating and mass sinks decays like a Gaussian with increasing radius $r$ provides expressions for short-time IPV anomalies produced by latent heating,

$$\tilde{P}'_{heat} = \cos\left(\pi \tilde{\theta}\right) \exp\left(-\tilde{r}^2\right), \tag{16}$$

and mass sinks,

$$\tilde{P}'_{mass} = \frac{1}{A} \tilde{\Pi} \sin\left(\pi \tilde{\theta}\right) \exp\left(-\tilde{r}^2\right), \tag{17}$$

where $\tilde{r} = r/L$ and $L$ is the horizontal length scale (Figure 7a-b). Because we will compute relative circulation strengths using a linearized inversion relation, our results will depend only on the relative magnitudes of IPV anomalies, and we have used solutions to Equation 15 at $\tilde{t} = 1/\pi$ for simplicity even though the short-time approximation ($\tilde{P}' << 1$) breaks down before then.

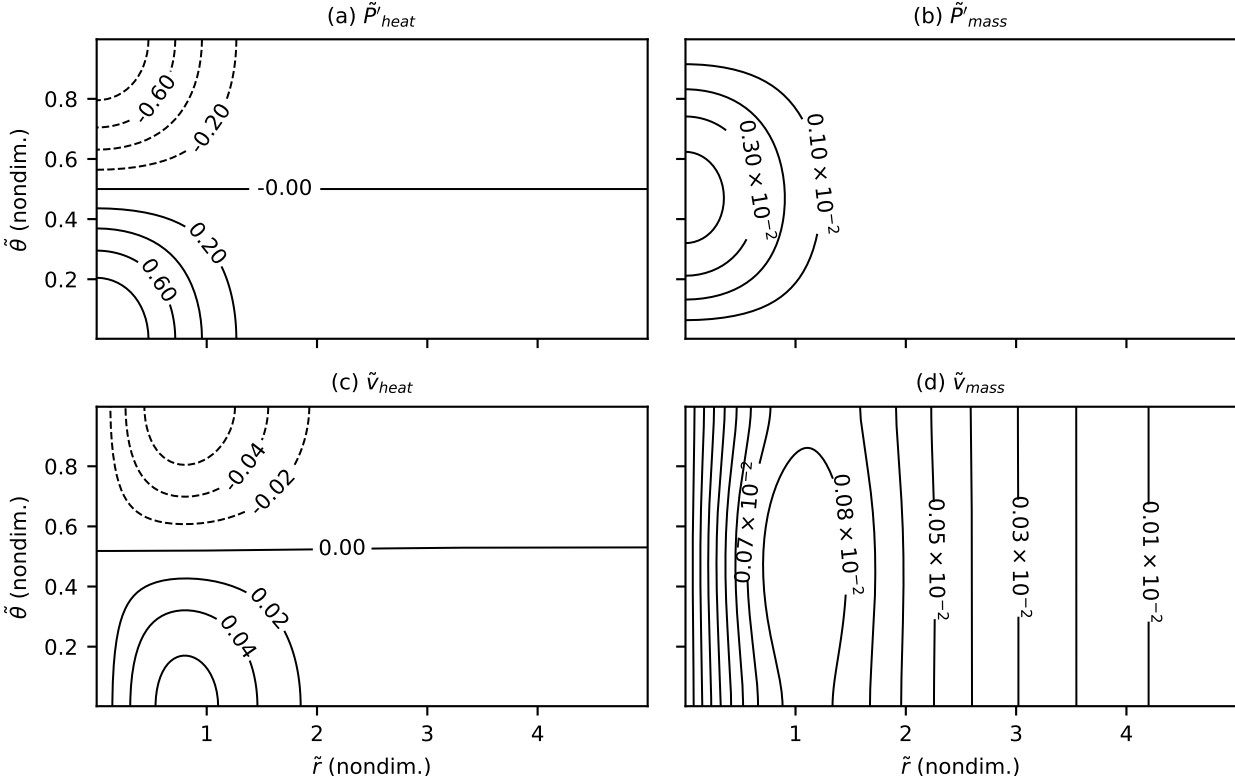

**Figure 7.** Short-time IPV anomalies produced by latent heating (a) and mass sinks (b), and azimuthal velocity fields induced by latent heating (c) and mass sinks (d) from a linearized inversion with $B = 1$.

Small-amplitude axisymmetric IPV anomalies and azimuthal velocities are approximately related by a linear elliptic equation (equation 32 of Hoskins et al., 1985):

$$\frac{\partial}{\partial r}\left(\frac{1}{r}\frac{\partial(rv)}{\partial r}\right) + \frac{f^2}{g\sigma_{ref}}\frac{\partial}{\partial \theta}\left(\left(\frac{\partial \Pi}{\partial p}\right)_{ref}^{-1}\frac{\partial v}{\partial \theta}\right) = \sigma_{ref}\frac{\partial P}{\partial r}. \tag{18}$$

Here, $v$ is the azimuthal velocity, $\sigma_{ref}$ is a reference-state isentropic density profile, and $(\partial \Pi/\partial p)_{ref}$ is the pressure derivative

of the Exner function in the reference state. Our earlier assumption that pressure varies linearly with potential temperature means that $\sigma_{ref}$ is a constant $\sigma_0 = g^{-1}f_p p_s/(\theta_t - \theta_s)$. Non-dimensionalizing by subsituting

$r \rightarrow L\tilde{r}$

$v \rightarrow fL\tilde{v}$

$p \rightarrow p_s\tilde{p},$

 in addition to earlier non-dimensionalizations, gives

$$\frac{\partial}{\partial \tilde{r}}\left(\frac{1}{\tilde{r}}\frac{\partial\left(\tilde{r}\tilde{v}\right)}{\partial \tilde{r}}\right) + \frac{1}{B}\frac{\partial}{\partial \tilde{\theta}}\left(\tilde{\Pi}_{p,ref}^{-1}\frac{\partial \tilde{v}}{\partial \tilde{\theta}}\right) = \frac{\partial \tilde{P}}{\partial \tilde{r}}. \tag{19}$$

as a non-dimensional inversion relation, with

$$\tilde{\Pi}_{p,ref} = \left(1 - f_p\tilde{\theta}\right)^{R/c_p - 1}. \tag{20}$$

B is a non-dimensional Burger number and can be written either as

$$B = \frac{L_D^2}{L^2}, \tag{21}$$

where

$$L_D = \frac{\left(g\sigma_0 R/p_s\right)^{1/2}\left(\theta_t - \theta_s\right)}{f} \tag{22}$$

is the Rossby deformation radius, or as

$$B = \frac{\left(\theta_t - \theta_s\right)^2}{\Delta\theta^2}, \tag{23}$$

where

$$\Delta\theta = \frac{fL}{\left(g\sigma_0 R/p_s\right)^{1/2}} \tag{24}$$

measures the vertical penetration of the inversion operator (Hoskins et al., 1985). Thus, we expect spatial cancellation during inversion to reduce the strength of latent-heating-induced circulations most when $\Delta\theta$ is large compared to $\theta_t - \theta_s$, or equivalently, when disturbances are large compared to a deformation radius.

To examine how spatial cancellation during inversion affects the relative strength of circulations induced by latent heating versus mass sinks, we use Equation 19 to invert $\tilde{P}'_{heat}$ (Equation 16) and $\tilde{P}'_{mass}$ (Equation 17) for azimuthal velocity fields $\tilde{v}_{heat}$ and $\tilde{v}_{mass}$, respectively. We perform inversions numerically on a domain with 1024 grid points between $\tilde{r} = 0$ and $\tilde{r} = 5$ and 256 grid points between $\tilde{\theta} = 0$ and $\tilde{\theta} = 1$, subject to boundary conditions $\tilde{v} = 0$ at the outer boundary and $\partial_{\tilde{\theta}}\tilde{v} = 0$ (no pressure perturbations) at the top and bottom boundaries. We assume that $\tilde{v}/\tilde{r}$ (or equivalently, the relative vorticity $\zeta_\theta$) tends

toward a constant as $r$ approaches 0. With $B = 1$, this produces a low-level cyclonic circulation and upper-level anticyclonic circulation for $\tilde{v}_{heat}$, and a vertically-extensive but much weaker cyclonic circulation for $\tilde{v}_{mass}$ (Figure 7c-d).

    We quantify the relative strength of circulations induced by latent heating and mass sinks by calculating latent-heating-to-mass sink ratios of maximum IPV anomaly, maximum azimuthal wind speed, and maximum near-surface azimuthal wind speed for inversions across a broad range of Burger number (Figure 8). The ratio of maximum IPV anomalies $(\max(\tilde{P}'_{heat})/\max(\tilde{P}'_{mass}))$

is independent of $B$ and close to the IPV tendency ratio $A$. The ratio of maximum azimuthal wind speeds $(\max|\tilde{v}_{heat}|/\max|\tilde{v}_{mass}|)$ approaches zero for small $B$ (horizontal length scales much larger than a deformation radius) because of vertical cancellation between opposite-signed regions of IPV anomalies from latent heating. For $B \sim 1$, however, vertical cancellation only moderately reduces the strength of circulations induced by latent heating, and the maximum wind speed induced by latent heating

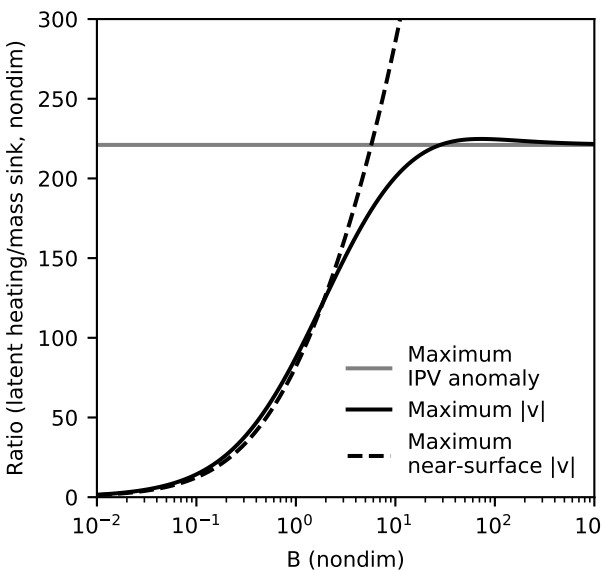

**Figure 8.** Ratios of maximum IPV anomalies (gray), azimuthal wind speeds (solid black), and near-surface azimuthal wind speeds (dashed black) produced by latent heating and the mass sink in inversions with varying Burger number $B = L_D^2/L^2$.

is larger than the maximum wind speed induced by mass sinks by a factor of about $A/2$. For larger $B$, the ratio of maximum wind speeds asymptotes to a value equal to the ratio of maximum IPV anomalies. Ratios of maximum near-surface wind speeds behave similarly at small and intermediate Burger number, but diverge at large Burger number because IPV anomalies from mass sinks are small near the surface.

These results, while likely to be sensitive to some degree to choices about the spatial structure of IPV anomalies, suggest that spatial cancellation during inversion between opposite-signed IPV anomalies is unlikely to significantly reduce the strength of circulations induced by latent heating compared to those induced by mass sinks. Typical length scales for precipitating regions of midlatitude disturbances are on the order of a deformation radius (e.g., extratropical cyclones) or smaller (e.g., across fronts). In these regimes ($B \gtrsim 1$), our inversions suggest that circulations induced by mass sinks are likely to be at least $\sim 100$ times weaker than circulations induced by latent heating. This ratio is somewhat smaller than the IPV tendency ratio $A$, but only by about a factor of 2.

Like our initial scale analysis of IPV tendencies, results from idealized calculations of the evolution of IPV in a precipitating column and related inversions are consistent with simulations showing no significant differences between runs without a mass sink and with the mass sink included. The idealized evolution and inversions also suggest that a tenfold exaggeration of mass sinks should allow them to induce circulations that are within about an order of magnitude of the strength of circulations induced by latent heating, which is at least plausibly consistent with simulations results showing a detectable effect from tenfold-exaggerated mass sinks. However, our theoretical analysis—which suggests that mass sinks should play a fairly unimportant role in both developing and mature disturbances—is difficult to square with the results of LY04, who demonstrated a

noticeable increase in TC intensity in a simulation of Hurricane Lilli with an unexaggerated precipitation mass sink included. In the following section, we attempt to reconcile our results with LY04 by testing the effect of mass sources and sinks in TC world simulations.

## 5 Impact of the Mass Sink in TC World Simulations

We generate TC worlds in the GFDL idealized spectral model by running simulations at T170 resolution with the sea surface temperature (SST) fixed to 300 K and the Coriolis parameter set to a value corresponding to 30 degrees latitude everywhere in the model domain. The choice of SST is not critical for TC genesis—simulations with globally-uniform SST can generate tropical cyclones at SSTs far above and below those found in TC genesis regions on modern Earth (e.g., Merlis et al., 2016)— but is likely to influence precipitation rates. As discussed below, our choice of SST generates precipitation rates comparable to LY04's Eta model simulations. We replace the gray radiative transfer scheme with fixed radiative cooling of 1.5 K day$^{-1}$ below the tropopause, defined as the lowest level with temperature less than 200 K, and temperature relaxation to 200 K on a 40 day time scale above the tropopause; and we run simulations with large-scale condensation but no convection scheme. Both changes are necessary to obtain TC worlds: simulations with gray radiation or the simplified Betts-Miller convection scheme (Frierson, 2007) struggle to maintain axisymmetric TC-like vortices. Additionally, we use $\nabla^4$ hyperdiffusion instead of $\nabla^8$ hyperdiffusion, increase the hyperdiffusion coefficients for vorticity and temperature by a factor of 5 (relative to our conventional T170 simulations), and increase the hyperdiffusion coefficient for divergence by a factor of 10. Changes to damping parameters are motivated by suggestions from Zhao et al. (2012) that stronger divergence damping may increase the number of TCs produced by global models. We initialize simulations with a quiescent isothermal atmosphere, run simulations at each planetary radius for 600 days without mass sources and sinks ($\gamma = 0$, "no MS"), and repeat the simulations with mass sources and sinks ($\gamma = 1$, "MS"). Statistics are calculated using instantaneous snapshots saved every 6 hours.

Simulations with this model configuration produce a population of small axisymmetric TC-like vortices (Figure 9a). At T170 resolution, which has an effective grid spacing of about 80 km at the equator and 60 km at midlatitudes, the vortices are only marginally resolved. Computational expense prevents us from easily running TC world simulations at higher resolution, so we instead test robustness to resolution by running additional simulations with the planetary radius reduced to one half and one quarter of Earth's radius. Smaller planetary radii produce simulations with better-resolved vortices but lower vortex count (Figure 9b,c). Quarter-Earth-radius simulations have an effective resolution of 20 km at the Equator, comparable to the 15 km resolution of Eta model simulations run by LY04. In-storm precipitation rates are likewise comparable in our TC world simulations and LY04's Eta model simulations: three-hour precipitation accumulations of 5-20 mm are widespread around TC cores, and accumulations of 20-60 mm occur over smaller regions (compare Figure 9d-f and LY04 Figure 10). Although we compare results in this section primarily to TC simulations analyzed by LY04, we note that the simulations of midlatitude frontal simulations described by Qiu et al. (1993) also use a horizontal resolution (80 km) within the range of our TC world simulations, and produce peak precipitation rates (10 mm/hour) comparable to our TC world simulations. Finally, we emphasize that the 20-to-40 km effective grid spacing of quarter- and half-Earth-radius simulations is comparable

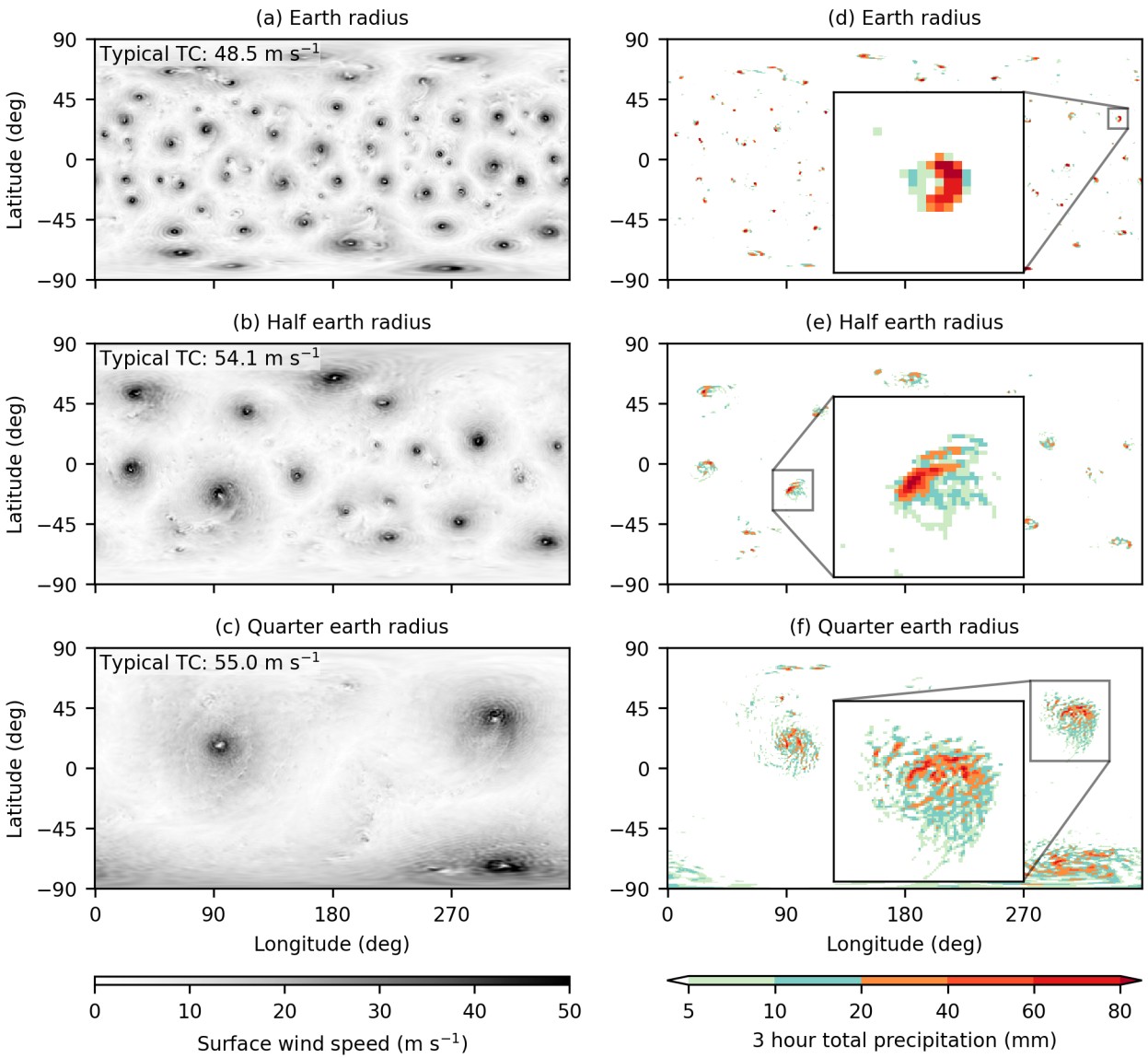

**Figure 9.** Snapshots of near-surface wind speed in no-MS TC world simulations with planetary radius equal to Earth's radius (a), half Earth's radius (b), and one-quarter Earth's radius (c), and total accumulated precipitation over the three-hour period following each snapshot (d-f). Values in the upper left corner of panels a-c show a measure of the maximum surface wind speed in a typical TC, defined as the maximum wind speed for the median TC (ranking all TCs at a given time by their maximum wind speed) averaged in time over the last 400 days of each simulation.

to resolutions typically used for global modeling of TCs (e.g. Zhao et al., 2009; Merlis et al., 2016; Chavas and Reed, 2019). Simulations at even higher resolution would be useful for assessing robustness, but computational expense prevents us from further decreasing the planetary radius (which requires a shorter time step), and TC size is moreover already comparable to the planetary surface area in the quarter-Earth-resolution simulations.

We analyze TC world simulations using a TC identification algorithm following Cronin and Chavas (2019). For each 6 hourly

snapshot, we mask regions with a surface pressure anomaly, measured relative to the zonal-mean surface pressure, less than -20 hPa. We tile the masked pressure field three times in longitude to permit identification of cyclones that cross longitudinal boundaries of the model output and label contiguous masked regions using an 8-connected component labeling algorithm. For each contiguous masked region, we define the latitude and longitude coordinates of the cyclone center as the surface-pressure-anomaly-weighted centroid of the 13 grid points with the lowest surface pressure anomaly. Unlike Cronin and Chavas (2019),

we retain masked regions with less than 13 grid points, in which case we include all grid points in the centroid calculation. We retain cyclones with centroids between 0 and 360 degrees longitude and -80 to 80 degrees latitude, the latter to avoid double-counting storms near the poles. We then define a central pressure anomaly for each cyclone by interpolating the surface pressure anomaly field to the cyclone centroid, and a maximum surface wind speed for each cyclone by finding the maximum surface wind speed within the region with a surface pressure anomaly less than -20 hPa. The maximum wind speed within the median

TC (ranking all TCs at a given time by their maximum wind speed) is highest in quarter-Earth-radius simulations, indicating that wind speeds in typical TCs increase at higher resolution (Figure 9a-c). By contrast, global-maximum surface wind speeds are highest in Earth-radius simulations simply because they contain more storms sampled from an intensity distribution with finite width.

Figure 10 shows time series of the central pressure anomaly, averaged across storms, and cyclone count for each TC world

simulation. The first identified cyclones appear after several tens of days, and the cyclone count increases rapidly before dropping and equilibrating to a statistically-steady value after 200 days. Cyclone count scales approximately linearly with planetary surface area, with about 50 cyclones at steady state in Earth-radius simulations, about 15 at steady state in half-Earth radius simulations, and 4-5 at steady state in quarter-Earth radius simulations. The inclusion or omission of mass sources and sinks has no clear effect on TC count. Time-average counts taken over four 100 day blocks between days 200 and 600 are

similar, with substantial overlap in ranges of values obtained for different 100 day blocks. Cyclone central pressure anomalies increase in magnitude over the first 100 days and also reach statistical steady states after 200 days. Central pressure anomalies averaged across storms increase slightly with increasing resolution, from about -45 hPa in Earth-radius simulations to about -50 hPa in quarter-Earth-radius simulations. Like cyclone counts, time-average central pressure anomalies are similar with and without mass sources and sinks, with significant overlap between the ranges of time averages taken over four 100 day

blocks between days 200 and 600. Additional quarter-Earth-radius experiments without changes to hyperdiffusion parameters produce TCs with slightly weaker central pressure anomalies, and likewise provide no evidence that precipitation mass sinks produce stronger TCs, though they suggest that weaker damping may allow mass sinks to increase TC count at the expense of TC intensity (Appendix E).

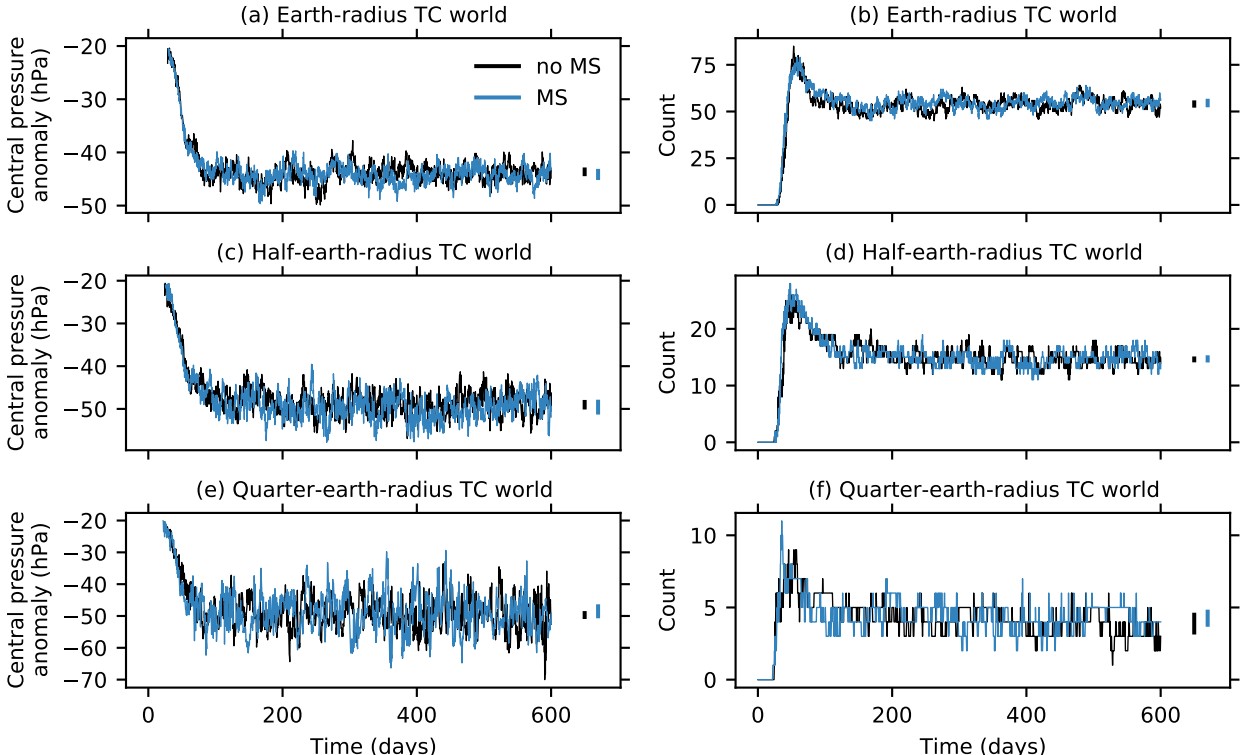

**Figure 10.** Time series of average TC central pressure anomalies (left) and TC count (right) in no-MS (black) and MS (blue) TC world simulations with planetary radius equal to Earth's radius (a-b), half Earth's radius (c-d), and one-quarter Earth's radius (e-f). Vertical bars at the end of time series indicate the range of time averages taken over four 100-day blocks between days 200 and 600.

So far, results from our TC world simulations (which show no detectable increase in TC intensity from including mass sources and sinks) are difficult to reconcile with LY04 (who found a clear intensity increase in simulations with mass sources and sinks). The disagreement cannot clearly be attributed to differences in resolution, as we detect no effect from including mass sources and sinks in quarter-Earth-radius simulations with resolutions and precipitation rates comparable to LY04's Eta model simulations. The disagreement is also unlikely to be the result of differences between rapidly-deepening versus mature cyclones: LY04's Eta model simulations produced very little deepening over the simulation period, and our TC world simulations show no clear difference between no-MS and MS simulations during the initial deepening period (Figure 10).

One potentially-important difference between our TC world simulations and the simulations presented by LY04 is that our simulations consider properties of a population of cyclones in equilibrium with their environment—and so with intensities potentially constrained by global budgets—while LY04 consider the transient evolution of a single TC in a short-term forecast. To test whether effects from mass sources and sinks are more readily apparent in transient simulations initialized from a common starting state, we restart two simulations from the end of the Earth-radius TC world simulation without mass sources

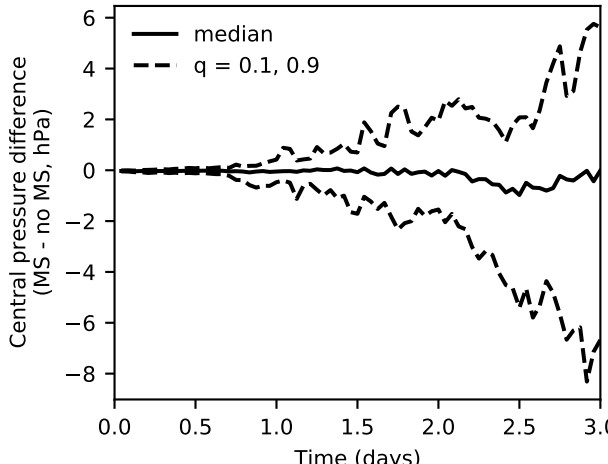

**Figure 11.** Transient evolution of pairwise differences in cyclone central pressure over the first 3 days of TC world restart experiments. The median and $q = 0.1$ and $= 0.9$ quantiles are calculated over the entire population of cyclones, and corresponding cyclones in experiments with and without the mass sink are matched as described in the text.

and sinks. In one simulation, we continue to omit mass sources and sinks; in the other simulation, we include them. We run these two simulations for three days and save instantaneous snapshots once every hour.

Once these two simulations are complete, we use our cyclone identification algorithm to calculate central pressure anomalies for each cyclone, and match corresponding TCs in the two simulations using the following algorithm. For each instantaneous snapshot, we consider a "source" set of cyclones from the simulation with fewer identified cyclones, and a "destination" set of cyclones from the other simulation. We match each cyclone in the source set with the nearest cyclone (measured by the great circle distance between centroids) in the destination set. This results in between 46 and 53 matched pairs for each snapshot. The matching is unique (i.e., there are not cases where multiple cyclones in the source set are matched to a single cyclone in the destination set) for the first 56 hours, and for 9 of the last 16 hours. In the remaining 7 of the last 16 hours, the number of unique cyclones matched from the destination set is one less than the number of cyclones in the source set, indicating that two cyclones in the source set were matched to the same cyclone in the destination set. Finally, we calculate the pairwise difference between central pressure anomalies for each matched pair.

Figure 11 shows time series of the median, 10th percentile, and 90th percentile pairwise difference between central pressure anomalies for each snapshot. The median central pressure difference varies little over the three days of simulation, with a slight decrease (less than 1 hPa) during the last 24 hours. However, the spread in 10th and 90th percentile central pressure differences increases substantially over the three days. The 10th percentile central pressure difference—which captures a pair of cyclones with a larger central pressure anomaly in the simulation with mass sources and sinks—decreases to about -8 hPa by day 3. The 90th percentile central pressure difference—which captures a pair of cyclones with a larger central pressure anomaly in the simulation without mass sources and sinks—departs from zero slightly more slowly, and reaches about 5 hPa by day 3.

This suggests one way to reconcile our results with LY04: including mass sources and sinks can have little effect on the intensity of an equilibrium population of TCs, and no obvious effect on the median intensity change in transient simulations of a large population of TCs, but still significantly alter the evolution of individual cyclones over time scales of 1-3 days. Our simulations do not show systematic TC deepening in simulations with mass sources and sinks included, but can produce substantial transient deepening for some individual cyclones, similar to results reported by LY04 for Hurricane Lili. To be clear,

we have not ruled out the possibility that model differences may bias our results relative to LY04. TCs in our simulations are relatively coarsely resolved, and obtaining TC-like vortices required running simulations without a convection scheme, both of which may affect our representation of TC dynamics. Nevertheless, our analysis of short-term TC world simulations provides a plausible bridge between our results for statistical equilibrium and the single-storm results of LY04.

## 6   Conclusions

In this paper, we presented idealized simulations and simple theoretical arguments that, together, suggest that water vapor mass sources and sinks are unlikely to have a significant impact on large-scale midlatitude circulations. In simulations with an idealized atmospheric general circulation model, statistics capturing midlatitude variability and extremes changed little when the model's dynamical core was modified to include mass sources and sinks in the mass conservation equation (Sections 2-3). This result held even in very warm climates with global-mean surface temperatures above 315 K and atmospheric water vapor

content much larger than modern Earth.

Our simulation results were consistent with analysis of an isentropic potential vorticity equation, which showed that PV sources and streamfunction anomalies associated with latent heating are typically larger than those associated with mass sinks by a non-dimensional factor of $\pi L_v / (c_p (\theta_t - \theta_s))$, where $L_v$ is the latent heat of vaporization of water vapor, $c_p$ is the heat capacity of dry air, and $\theta_t - \theta_s$ is the potential temperature range over which condensation occurs. For troposphere-deep ascent

in Earth's atmosphere ($\theta_t - \theta_s \approx 40$ K), this non-dimensional number is about 200, suggesting that latent heating should produce PV sources and sinks about two orders of magnitude larger than those produced by mass sources and sinks. Warmer climates will produce storms with much heavier precipitation rates, but the ratio of PV sources for latent heating versus mass sinks will remain similar. Re-purposing this result for other planets with other condensible species is straightforward: for troposphere-deep ascent on Titan ($\theta_t - \theta_s \approx 20$ K, e.g., Mitchell and Lora (2016)), where methane is the main condensible species ($L_v \approx$

$5 \times 10^5$ J kg$^{-1}$) and the atmosphere is mostly nitrogen ($c_p \approx 10^3$ J kg$^{-1}$ K$^{-1}$), $\pi L_v / (c_p (\theta_t - \theta_s)) \approx 80$, suggesting that PV sources and sinks from troposphere-deep latent heating should be about a factor of 80 larger than those from corresponding mass sinks.

Our simulation results stand in contrast to previous work that found that the precipitation mass sink could enhance precipitation rates in mesoscale weather systems by about 10% (Qiu et al., 1993) and increase peak surface wind speeds in hurricanes

by 5 to 15 knots (Lackmann and Yablonsky, 2004). Based on TC world simulations, we suggest that one way to reconcile our results with previous work is by recognizing that including mass sources and sinks can substantially alter the evolution of

individual weather systems on time scales of days without having a systematic effect on populations of many individual storms (Section 5).

Our results reveal no urgent need to revisit existing literature on links between climate and statistical properties of midlatitude storm tracks. Our results directly confirm that a reduction in midlatitude EKE with warming at statistical equilibrium on an aquaplanet, documented in previous work (O'Gorman and Schneider, 2008a), remains virtually unchanged in simulations that include mass sources and sinks. Our work did not examine Lagrangian statistics that measure e.g. the size, intensity, or frequency of midlatitude cyclones themselves (Pfahl et al., 2015), but our simulation results and analysis of PV theory suggest that those are also unlikely to depend significantly on the inclusion of mass sources and sinks. Our work also did not explore climates in which water vapor becomes extremely non-dilute and the character of large-scale flows becomes very different (Pierrehumbert and Ding, 2016). Including mass sources and sinks for condensible species certainly broadens the range of applications for large-scale atmospheric models, and they are included in many models that target simulations of planetary atmospheres (Pierrehumbert and Ding, 2016; Forget et al., 2017; Neary and Daerden, 2018; Chow et al., 2019). The Martian general circulation in particular includes a planetary-scale meridional component that is driven by seasonal condensation of carbon dioxide at the winter pole and clearly influenced by changes in surface pressure produced by carbon dioxide mass sinks (Pollack et al., 1981, 1990; Haberle et al., 1993; Chow et al., 2019). However, omitting mass sources and sinks seems a reasonable simplification for simulations of midlatitude eddies in Earth's atmosphere, even in climates substantially warmer than the modern.

At the same time, some of our results—specifically, our short-term TC world simulations—do indicate that mass sources and sinks can have appreciable effects on the transient evolution of strongly-precipitating weather systems, as was suggested by LY04. This raises the possibility that short-term forecasts may benefit from correctly including mass sources and sinks. In our view, however, it remains unclear whether mass sources and sinks are "missing physics" with the potential to provide systematic improvements to forecast skill, or whether their inclusion simply provides a small perturbation that eventually alters the trajectory of a chaotic dynamical system. This issue could be addressed in future work using simulations similar to our TC world restart experiments, or additional simulations of TC case studies, with a focus on comparing how trajectories diverge in pairs of simulations with and without mass sources and sinks and in pairs of simulations distinguished by small random perturbations to initial conditions.

Overall, our study offers evidence, from both simulations and theory, suggesting that mass sources from evaporation and mass sinks from precipitation are unlikely to have a significant effect on large-scale midlatitude variability. Our results do not necessarily preclude the possibility that mass sources and sinks may have a more important role on the transient evolution of strongly precipitating mesoscale weather systems, including midlatitude fronts (Qiu et al., 1993) and tropical cyclones (Lackmann and Yablonsky, 2004), as illustrated for the transient evolution of individual tropical cyclones in Section 5. Considering the role of precipitation mass sinks in an idealized frontogenesis problem could help to clarify their role in the dynamics of mesoscale midlatitude weather systems. Future work could also examine the effects of mass sources and sinks on the transient evolution and equilibrium statistics of TCs in idealized cloud-system-resolving simulations, which produce TC worlds when run on an f-plane (e.g., Merlis and Held, 2019). The role of mass sinks in the formation, organization, and vigor of

individual convective cells could be explored through simulations with cloud-resolving models based on fully-compressible non-hydrostatic equations that carefully represent the effects of precipitation fallout on mass, energy, entropy, and momentum budgets (e.g., Ooyama, 2001; Bott, 2008). Finally, models that include a more sophisticated treatment of condensed water (including condensate loading and prognostic hydrometeor fallout) could explore whether processes neglected in our pseudo-diabatic model enhance the dynamical effects of precipitation mass sinks. All are potentially interesting directions for future research.

*Code and data availability.* The GFDL idealized moist general circulation model is publicly available at https://www.gfdl.noaa.gov/idealized-moist-spectral-atmospheric-model-quickstart. Modifications to the model source code, simulation input files and postprocessed output, and scripts for reproducing figures are available at https://doi.org/10.5281/zenodo.8407812 (Abbott and O'Gorman, 2023).

## Appendix A: Model changes

Modifications to the idealized GCM couple sources and sinks of moisture calculated by physics parameterizations to pressure tendencies calculated by the dynamical core while retaining the ability to run simulations into equilibrium across a wide range of climates. We emphasize that the changes are not intended to produce a model fully valid for simulations with strongly non-dilute water vapor, and the model retains approximations—like the use of a humidity-independent heat capacity—that become increasingly inaccurate as the moisture content of the atmosphere increases. Rather, the goal of the changes documented here is to add a representation of the "precipitation mass sink", or pressure tendencies resulting from sources and sinks of water vapor, while making relatively few other changes to the model equations.

Incorporating the precipitation mass sink is done in three steps. First, the pressure tendency calculation in the dynamical core is modified to include a term for a water vapor mass source (Appendix A1). Second, mass sources are calculated from specific humidity tendencies produced by the convection, large-scale condensation, and surface flux schemes and passed to the modified pressure tendency calculation (Appendix A2). Finally, a mass fixer is added to the model dynamical core to correct numerical errors in the conservation of dry mass introduced by the inclusion of sources and sinks of moist mass (Appendix A3).

In addition to changes made to incorporate the precipitation mass sink, we make some minor changes to lapse rate calculations in the convection scheme (Appendix A4) to improve consistency with the treatment of thermodynamics and moisture in other parts of the GCM.

### A1  Pressure tendency calculations

Pressure tendency calculations are derived from the mass continuity equation. For a parcel with dry mass density $\rho_d$, total density $\rho$, water vapor mixing ratio $r_v$, and spatial dimensions $\delta x$, $\delta y$, and $\delta z$, mass conservation requires

$$\frac{\mathrm{D}}{\mathrm{D}t}\rho\,\delta x\delta y\delta z = \frac{\mathrm{D}}{\mathrm{D}t}(1+r_v)\rho_d\,\delta x\delta y\delta z = \rho_d\,\delta x\delta y\delta z\frac{\mathrm{D}r_v}{\mathrm{D}t}, \tag{A1}$$

where the second equality follows from the fact that the parcel conserves dry mass. From hydrostatic balance,

$$\delta x \delta y \delta z = -\frac{\delta x \delta y \delta p}{\rho g}, \tag{A2}$$

where $p$ is pressure and $g$ is the acceleration from gravity, giving

$$\frac{1}{\delta x \delta y \delta p} \frac{\mathrm{D}}{\mathrm{D}t} \delta x \delta y \delta p = \frac{\rho_d}{\rho} \frac{\mathrm{D}r_v}{\mathrm{D}t}. \tag{A3}$$

Replacing $\mathrm{D}\delta x/\mathrm{D}t$ with $\delta u$ and $\delta u/\delta x$ with $\partial_x u$, and making similar substitutions for $y$ and $p$, gives

$$\nabla \cdot \mathbf{v} + \frac{\partial \omega}{\partial p} = \frac{\rho_d}{\rho} \frac{\mathrm{D}r_v}{\mathrm{D}t}, \tag{A4}$$

where $\nabla = (\partial/\partial x, \partial/\partial y)$ taken at constant $p$, $\mathbf{v} = (u, v)$ is the horizontal velocity, and $\omega$ is the vertical velocity in pressure coordinates. Replacing the mixing ratio with specific humidity $q_v$ gives

$$\nabla \cdot \mathbf{v} + \frac{\partial \omega}{\partial p} = \frac{1}{1 - q_v} \frac{\mathrm{D}q_v}{\mathrm{D}t}. \tag{A5}$$

Within the model, we replace $\partial_p \omega = -\nabla \cdot \mathbf{v}$ in pressure tendency calculations with

$$\frac{\partial \omega}{\partial p} = -\nabla \cdot \mathbf{v} + \gamma \frac{1}{1 - q_v} \frac{\mathrm{D}q_v}{\mathrm{D}t}. \tag{A6}$$

Setting an appropriate value of $\gamma$ allows simulations without mass sources and sinks ($\gamma = 0$), with mass sources and sinks ($\gamma = 1$), and with exaggerated mass sources and sinks ($\gamma > 1$). In pressure coordinates, integrating downward from $\omega = 0$ at

635 $p = 0$ gives

$$\omega(p) = -\nabla \cdot \int_0^p \mathbf{v} \mathrm{d}p' + \int_0^p \gamma \left( \frac{1}{1 - q_v} \frac{\mathrm{D}q_v}{\mathrm{D}t} \right) \mathrm{d}p'. \tag{A7}$$

The second term is the contribution to the pressure tendency $\omega(p)$ from the source of water vapor described by $\mathrm{D}q_v/\mathrm{D}t$. Within the sigma-coordinate dynamical core, the modified expression for $\partial_p \omega$ (Equation A6) is used by existing (un-modified) code to calculate surface pressure tendencies and sigma-coordinate vertical velocities that incorporate contributions from mass sources

and sinks.

## A2 Mass sources and sinks from moist physics

We include mass sources and sinks from the large-scale condensation scheme, the convection scheme, and surface evaporation.

The large-scale condensation scheme produces specific humidity tendencies from condensation and evaporation, and we add resulting mass sources and sinks to the pressure tendency calculation in the dynamical core following Equation A6.

The convection scheme also produces specific humidity tendencies, but without distinguishing between tendencies from condensation and evaporation (which add and remove mass from parcels) and tendencies from sub-grid-scale transport (which conserves mass following parcels). We assume that specific humidity tendencies are entirely from parcel mass sources and

sinks in columns where the convection scheme is running in "deep convection" mode (which produces some surface precipitation) and entirely from sub-grid transport in columns where the convection scheme is running in "shallow convection" mode (which produces no surface precipitation). Thus, we add mass sources and sinks to the pressure tendency calculation following Equation A6 for columns and times with parameterized deep convection but not parameterized shallow convection.

The surface flux scheme provides the mass flux of water vapor from surface evaporation directly, and we add a mass source to the lowest model level based on the mass flux from surface evaporation.

### A3    Mass fixer

The implementation of water vapor mass sources and sinks produces a model that only conserves dry mass up to first order in the size of specific humidity increments, and the resulting numerical errors result in a gradual downward drift of dry atmospheric mass. This downward drift of dry mass occurs even if moist mass sources and sinks from evaporation and precipitation are exactly in balance (see Equation A12) and prevents stable simulations if left uncorrected. We compensate for unphysical reductions in dry mass produced by numerical errors by using a mass fixer that ensures that the model conserves a fixed inventory of dry mass while leaving an additional component of atmospheric mass free to adjust in response to moist mass sources and sinks.

We can show that the numerical errors enter at second order and tend to reduce dry atmospheric mass by considering a parcel with mass $m$ and specific humidity $q_v$ before and after an update that alters specific humidity an amount $dq$ and mass by an amount $dm$. Before the update, the parcel mass is related to specific humidity and dry mass $m_d$ by

$$m = \frac{m_d}{1 - q_v}. \tag{A8}$$

Because mass changes are computed assuming that parcels conserve dry mass (see Appendix A1), the parcel mass after the update is given by

$$m' = m + dm = m + d\left(\frac{m_d}{1 - q_v}\right) = m + \frac{m_d}{(1 - q_v)^2}dq_v = \frac{m}{1 - q_v}dq_v, \tag{A9}$$

and the specific humidity after the update is given by

$$q_v' = q_v + dq_v. \tag{A10}$$

The implied dry mass of the parcel after the update (which may differ from the parcel's initial dry mass because of numerical errors) is given by

$$m' = \frac{m_d'}{1 - q_v'}. \tag{A11}$$

Combining Equations A8-A11 and solving for $m_d'$ gives

$$m_d' = m_d - \frac{m}{1 - q_v}dq_v^2. \tag{A12}$$

Dry mass conservation requires $m_d' = m_d$, but instead $m_d'$ is reduced relative to $m_d$ by an amount that scales with $dq_v^2$.

To obtain stable simulations, we include a mass fixer to enforce conservation of a fixed inventory of dry atmospheric mass. Before each time step, the fixer calculates the current atmospheric dry mass inventory

$$M_d = M - \gamma M_w, \tag{A13}$$

where $M$ is the total atmospheric mass, $M_w$ is the mass of water vapor, and $\gamma$ is the factor used to rescale the water vapor mass source in the pressure tendency calculation. After each time step, the fixer calculates the new dry atmospheric mass

$$M_d' = M' - \gamma M_w', \tag{A14}$$

where $M'$ and $M_w'$ are total and water vapor mass after the time step but before applying corrections. Finally, the fixer multiplies surface pressure by a global multiplicative constant and divides specific humidity by the same constant to set $M_d'$ to the dry mass inventory $M_d$ calculated before the time step while leaving $M_w'$ unchanged. Because our model uses sigma coordinates, adjusting the total atmospheric mass by multiplying surface pressure by a single number in all columns changes the mass thickness of each level by an amount proportional to that level's mass thickness. This procedure does not add dry mass in precisely the places where it is lost due to numerical errors, which we view as acceptable because changes in dry mass (proportional to $\mathrm{d}q_v^2$) are locally small compared to changes in moist mass (proportional to $\mathrm{d}q_v$), and the mass fixer is used primarily to allow stable long-term simulations that correctly conserve total dry mass.

**A4  Lapse rate calculations**

In the default version of the model, expressions for lapse rates used to compute reference adiabats used by the simplified Betts-Miller convection scheme differ from lapse rates derived from the governing equations for the resolved flow. In our simulations, we use a modified convection scheme where lapse rates are computed consistently with equations for resolved flow. The inconsistencies corrected by the modified convection scheme are not introduced by the addition of mass sources and sinks, but we document them here for completeness.

Expressions for adiabatic lapse rates in unsaturated and saturated air can be derived from the temperature equation in the model dynamical core,

$$\frac{\mathrm{D}T}{\mathrm{D}t} = \frac{\kappa T_v}{p}\frac{\mathrm{D}p}{\mathrm{D}t} + \frac{Q}{c_p}, \tag{A15}$$

the approximate equation used by the model to compute virtual temperature,

$$T_v = T\left(1 + \epsilon q_v\right) \tag{A16}$$

with $\epsilon = R_v/R_d - 1$, the relationship between specific humidity and temperature changes in the large-scale condensation scheme,

$$\left(\frac{\mathrm{D}T}{\mathrm{D}t}\right)_{\text{condensation}} = -\frac{L_v}{c_p}\frac{\mathrm{D}q_v}{\mathrm{D}t}, \tag{A17}$$

the Clausius-Clapeyron equation used to compute saturation vapor pressure,

$$e^*(T) = e^*(T_0) \exp\left(-\frac{L_v}{R_v}\left(\frac{1}{T} - \frac{1}{T_0}\right)\right), \tag{A18}$$

the relationship between saturation vapor pressure and saturation specific humidity

$$q_v^*(T,p) = \frac{R_d e^*(T)}{R_v p - (R_v - R_d) e^*(T)}, \tag{A19}$$

and the relationship between saturation vapor pressure and saturation mixing ratio

$$r_v^*(T,p) = \frac{R_d e^*(T)}{R_v (p - e^*(T))}, \tag{A20}$$

In these equations, $T$ is temperature, $\kappa = R_d/c_p$ is the ratio of the specific gas constant and heat capacity of dry air, $Q$ is a heat source, $L_v$ is the latent heat of vaporization, $q_v$ is specific humidity, $R_v$ is the specific gas constant for water vapor, $e^*$ is saturation vapor pressure, and $q_v^*$ is saturation specific humidity.

For an unsaturated adiabat, $Q/c_p = 0$ and $q_v$ is conserved, giving

$$\frac{\mathrm{D}T}{\mathrm{D}\log p} = \kappa T_v \tag{A21}$$

and

$$T(p) = T(p_0)\left(\frac{p}{p_0}\right)^{\kappa(1+\epsilon q_v)}. \tag{A22}$$

These expressions replace

$$\frac{\mathrm{D}T}{\mathrm{D}\log p} = \kappa T \tag{A23}$$

and

$$T(p) = T(p_0)\left(\frac{p}{p_0}\right)^{\kappa} \tag{A24}$$

in the calculation of reference temperature profiles below the LCL in the modified version of the convection scheme.

For a saturated adiabat, $Q/c_p = \left(\frac{\mathrm{D}T}{\mathrm{D}t}\right)_{\text{condensation}}$ and $q_v = q_v^*$, and Equations A15-A19 give

$$\frac{\mathrm{D}T}{\mathrm{D}\log p} = \frac{R_d T_v - L_v \frac{\partial q_v^*}{\partial \log p}}{c_p + L_v \frac{\partial q^*}{\partial T}}, \tag{A25}$$

$$\frac{\partial q_v^*}{\partial \log p} = -\alpha r_v^*, \tag{A26}$$

and

$$\frac{\partial q_v^*}{\partial T} = \alpha \frac{L_v r_v^*}{R_v T^2}, \tag{A27}$$

with

$$\alpha = \frac{p(p - e^*)}{\left(p - \left(1 - \frac{R_d}{R_v}\right)e^*\right)^2}. \tag{A28}$$

Combining gives

$$\frac{\mathrm{D}T}{\mathrm{D}\log p} = \frac{R_d T_v + \alpha L_v r_v^*}{c_p + \alpha \frac{L_v^2 r_v^*}{R_v T^2}}. \tag{A29}$$

This expression replaces

$$\frac{\mathrm{D}T}{\mathrm{D}\log p} = \frac{R_d T + L_v r_v^*}{c_p + \frac{L_v^2 r_v^*}{R_v T^2}}. \tag{A30}$$

in the calculation of reference temperature profiles above the LCL in the modified version of the convection scheme.

We note that the calculation of the LCL itself—done using a lookup table—still assumes that $\mathrm{D}T/\mathrm{D}\log p = \kappa T$ below the LCL. Changing the LCL calculation to use $\mathrm{D}T/\mathrm{D}\log p = \kappa T_v$ would have required the current one-dimensional lookup table that estimates LCL pressure to be replaced by a two-dimensional lookup table.

## A5   Model verification

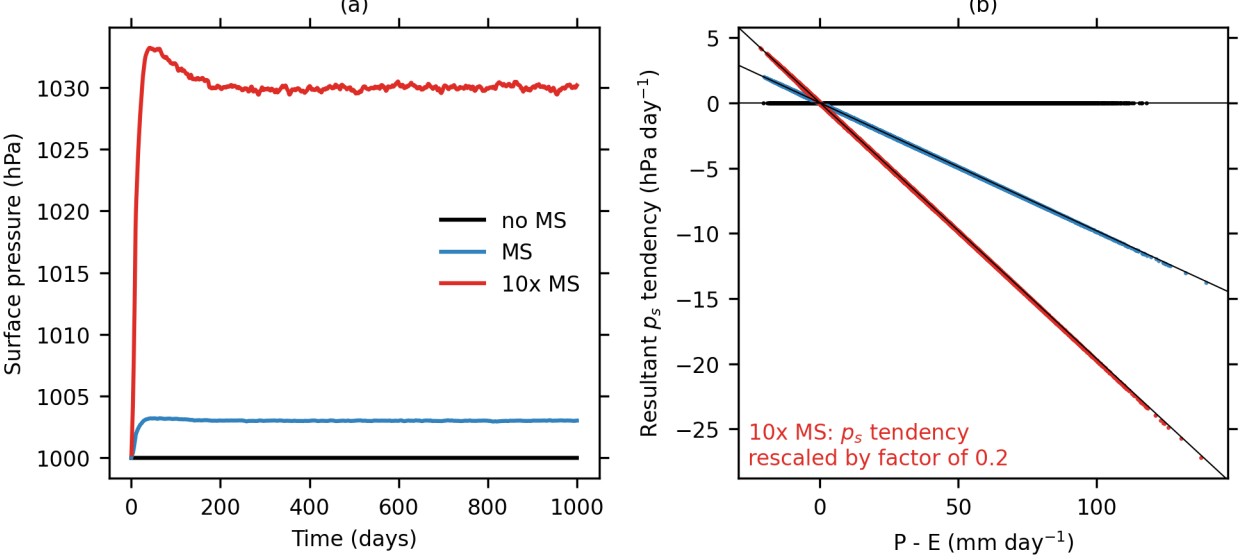

**Figure A1.** Time series of global-mean surface pressure (a) and relationship between precipitation minus evaporation ($P - E$) and the component of the surface pressure tendency from the mass sink (b) in T42 simulations at $\alpha = 1$. Points in panel (b) are for different grid boxes and instantaneous times at all latitudes, and diagonal lines in panel (b) show the expected relationship based on hydrostatic balance (see text). Surface pressure tendencies for the 10x-MS simulation are rescaled by a factor of 0.2 in panel (b) to make surface pressure tendencies for MS and no-MS simulations more easily visible.

Figure A1 shows time series of total atmospheric mass and the relationship between instantaneous precipitation minus evaporation and the surface pressure tendency from mass sources and sinks from global (not TC world) simulations run at T42 resolution for the control climate ($\alpha = 1$) without mass sources and sinks ($\gamma = 0$, "no MS"), with mass sources and sinks ($\gamma = 1$, "MS"), and with mass sources and sinks exaggerated by a factor of 10 ($\gamma = 10$, "10x MS"). All simulations are started from a dry atmosphere with the same initial mass. The total atmospheric mass (as measured by the global-mean surface pres-
sure) remains constant in the simulation without mass sources and sinks, increases slightly in the simulation with mass sources and sinks, and increases by about a factor of 10 more with tenfold-exaggerated mass sources and sinks (Fig A1a). After 200-300 days, the atmospheric mass reaches a steady state with no appreciable drift. The surface pressure tendency corresponding to mass sources and sinks is expected from hydrostatic balance to scale with precipitation minus evaporation ($P - E$) as

$$\left( \frac{\partial p_s}{\partial t} \right)_{\text{mass sources and sinks}} = -\gamma g \left( P - E \right) \tag{A31}$$

with a proportionality factor set by $\gamma$ and the acceleration from gravity $g$, and this is confirmed in Figure A1b.

## Appendix B: Relationship between pseudoadiabatic and non-pseudoadiabatic continuity equations

     Equation A1 assumes, consistent with the pseudoadiabatic limit used in our model, that the atmosphere contains only dry air and water vapor. In a more realistic limit, where condensed water is present and parcels gain and lose mass only through the vertical sedimentation of condensed water, mass conservation requires

$$755 \quad \frac{\mathrm{D}}{\mathrm{D}t} \rho \delta x \delta y \delta z = F^+ \delta x \delta y - F^- \delta x \delta y \tag{B1}$$

where $F^+$ and $F^-$ are the sedimentation mass fluxes of condensed water (dimensions of mass per unit area per unit time, defined as positive downward) through the top and bottom of the parcel. Using hydrostatic balance to replace $\delta z$ with $\delta p$ gives

$$\frac{1}{\delta x \delta y \delta p} \frac{\mathrm{D}}{\mathrm{D}t} \delta x \delta y \delta p = -\frac{g}{\delta p} \left( F^+ - F^- \right). \tag{B2}$$

Replacing $\mathrm{D}\delta x / \mathrm{D}t$ with $\delta u$, making similar substitutions for $y$ and $p$, and replacing $(F^+ - F^-)/\delta p$ with $\partial F / \partial p$ gives

$$760 \quad \nabla \cdot \mathbf{v} + \frac{\partial \omega}{\partial p} = -g \frac{\partial F}{\partial p}. \tag{B3}$$

     The pseudoadiabatic version of the continuity equation (Equation A4) can be recovered from Equation B3 by assuming that time scales for precipitation formation and fallout are infinitely fast. In this limit, the sedimentation flux at pressure level $p$ must be equal to the vertically-integrated net condensation at all higher levels, because any condensed water produced at higher levels will fall through lower levels within an infinitesimally short time. Because the net condensation rate per unit
volume is given by $-\rho_d \mathrm{D}r_v / \mathrm{D}t$, the sedimentation flux must be

$$F(p) = -\int_0^p \rho_d \frac{\mathrm{D}r_v}{\mathrm{D}t} \frac{\mathrm{d}p'}{\rho g}, \tag{B4}$$

and differentiating with respect to $p$ gives

$$-g\frac{\partial F}{\partial p} = \frac{\rho_d}{\rho}\frac{\mathrm{D}r_v}{\mathrm{D}t}. \tag{B5}$$

This recovers Equation A4 from the right-hand side of Equation B3 under the assumption of infinitely fast precipitation for-
mation and fallout.

## Appendix C: Continuity equation with mass sources and sinks in potential temperature coordinates

Beginning from Equation A3 and using the isentropic density $\sigma$ to substitute $\delta p = -g\sigma\delta\theta$ gives

$$\frac{1}{\sigma\delta x\delta y\delta\theta}\frac{\mathrm{D}}{\mathrm{D}t}\sigma\delta x\delta y\delta\theta = \frac{\rho_d}{\rho}\frac{\mathrm{D}r_v}{\mathrm{D}t}. \tag{C1}$$

Recall that the material derivative in potential temperature coordinates is

$$\frac{\mathrm{D}}{\mathrm{D}t} = \frac{\partial}{\partial t} + \mathbf{v}\cdot\nabla_\theta + \dot\theta\frac{\partial}{\partial\theta}, \tag{C2}$$

where $\mathbf{v}$ is the horizontal velocity and and $\nabla_\theta$ is the gradient along potential temperature surfaces. Expanding the left-hand-
side derivative in Equation D1, making substitutions similar to those used to derive Equation A4, using the definition of the
material derivative in potential temperature coordinates, and replacing $r_v$ with $q_v$ gives

$$\frac{\partial\sigma}{\partial t} + \nabla_\theta\cdot(\sigma\mathbf{v}) + \frac{\partial}{\partial\theta}\left(\sigma\dot\theta\right) = \frac{\sigma}{1 - q_v}\frac{\mathrm{D}q_v}{\mathrm{D}t}. \tag{C3}$$

## Appendix D: Relationship between pseudoadiabatic and non-pseudoadiabatic IPV equations

Beginning from Equation B2 and using the isentropic density $\sigma$ to substitute $\delta p = -g\sigma\delta\theta$ gives

$$\frac{1}{\sigma\delta x\delta y\delta\theta}\frac{\mathrm{D}}{\mathrm{D}t}\sigma\delta x\delta y\delta\theta = \frac{1}{\sigma\delta\theta}\left(F^+ - F^-\right). \tag{D1}$$

Following steps similar to those outlined in Appendix C then gives

$$\frac{\partial\sigma}{\partial t} + \nabla_\theta\cdot(\sigma\mathbf{v}) + \frac{\partial}{\partial\theta}\left(\sigma\dot\theta\right) = \frac{\partial F}{\partial\theta} \tag{D2}$$

as a non-pseudoadiabatic version of the continuity equation in potential temperature coordinates. This is identical to Equation
2 used to derive the pseudoadiabatic IPV equation, but with $\sigma\dot q_v$ replaced by $\partial F/\partial\theta$. Following the steps described in Section
4.1 with the modified continuity equation gives

$$\frac{\mathrm{D}P}{\mathrm{D}t} = P\left(\frac{\partial\dot\theta}{\partial\theta} - \frac{1}{\sigma}\frac{\partial F}{\partial\theta}\right) + \sigma^{-1}\mathbf{k}\cdot\left(\frac{\partial\mathbf{v}}{\partial\theta}\times\nabla_\theta\dot\theta\right) \tag{D3}$$

as a non-pseudoadiabatic version of the IPV equations. Similar to Equation 21 of Schubert et al. (2001), this version of the IPV
equation includes a source term from mass sinks that involves the divergence of the hydrometeor sedimentation mass flux.

The pseudoadiabatic IPV equation (Equation 6) derived in Section 4 can be recovered from the Equation D3 by assuming that precipitation formation and fallout is infinitely fast. Under this assumption, the divergence of the hydrometeor sedimentation mass flux must be given by Equation B5. With the approximation $1 - q_v \approx 1$, used in the derivation in Section 4, Equation B5 becomes

$$-g\frac{\partial F}{\partial p} = \dot{q_v}. \tag{D4}$$

Using $\sigma = -g^{-1}\partial p/\partial \theta$ gives

$$-\sigma g\frac{\partial F}{\partial p} = \frac{\partial F}{\partial \theta} = \sigma \dot{q_v}. \tag{D5}$$

Substituting $\partial F/\partial \theta = \sigma \dot{q_v}$ into the non-pseudoadiabatic IPV equation (Equation D3) recovers the pseudoadiabatic IPV equation (Equation 6) used in Section 4.

## Appendix E:  Impact of precipitation mass sinks in TC world simulations without hyperdiffusion changes

The use of $\nabla^4$ hyperdiffusion and increased hyperdiffusion coefficients is not essential for producing tropical cyclones in TC world simulations. Quarter-Earth-radius TC world simulations with "default" hyperdiffusion parameters ($\nabla^8$ hyperdiffusion, and no increases to hyperdiffusion coefficients relative to conventional T170 simulations) produce TCs with slightly weaker central pressure anomalies compared to quarter-Earth radius simulations with modified hyperdiffusion (compare Figure E1a with Figure 10e), and provide no evidence for systematic TC deepening when mass sinks are added (Figure E1a). Unlike simulations with modified hyperdiffusion, simulations with default hyperdiffusion parameters suggest that including mass sinks may produce simulations with slightly higher TC counts and slightly weaker average central pressure anomalies (compare averages shown by vertical bars in Figure E1), possibly indicating that mass sinks help maintain relatively weak TCs. This effect, if real, would be interesting and worthy of further investigation, but is distinct from the focus of our work, which is on whether mass sinks produce systematically stronger TCs.

*Author contributions.* **Conceptualization**: Abbott and O'Gorman. **Model source modification and simulations**: Abbott. **Analysis and interpretation**: Abbott and O'Gorman. **Writing—preparation of original draft**: Abbott. **Writing—review and editing**: Abbott and O'Gorman.

*Competing interests.*  The authors declare that they have no conflicts of interest.

*Acknowledgements.*  Tristan H. Abbott acknowledges support from the John W. Jarve (1978) Seed Fund for Science Innovation, and Paul A. O'Gorman acknowledges support from NSF AGS 2031472.

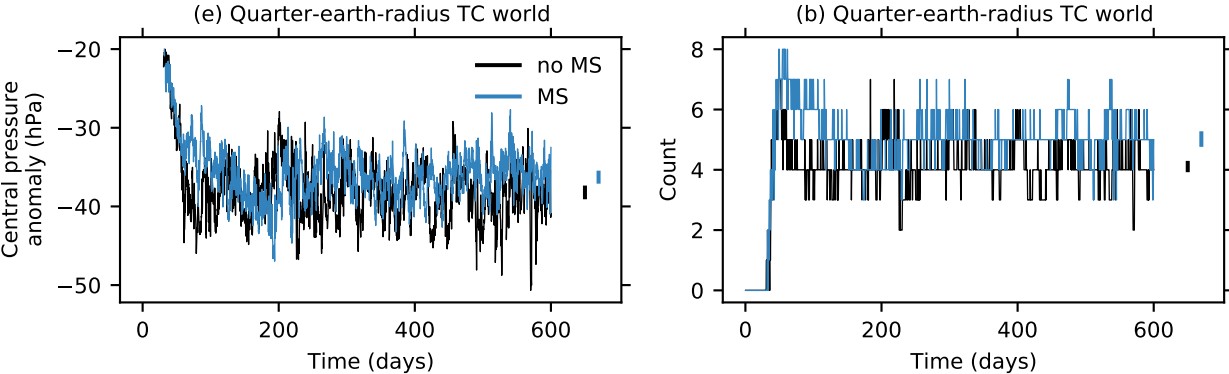

**Figure E1.** Time series of average TC central pressure anomalies (a) and TC count (b) in quarter-Earth-radius TC world simulations without hyperdiffusion increases. Vertical bars at the end of time series indicate the range of time averages taken over four 100-day blocks between days 200 and 600.

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
