# Peer review of "Impact of Precipitation Mass Sinks on Midlatitude Storms in Idealized GCM Simulations over a Wide Range of Climates"

_EGUsphere, 2023_

## Author Response (AR1)

Note: this document contains reviewer comments in black text and our responses in blue, with references to changes in the revised manuscript highlighted in red. Line numbers in our responses refer to the tracked-changes version of the manuscript. Our responses largely follow those posted in the public online discussion, but we have edited them where appropriate to refer to actual rather than planned changes.

**RC1**

Summary: This is an interesting and thorough paper investigating the importance of a physical process that is typically neglected, the "precipitation mass sink/evaporation mass source". The authors explore whether neglect of these sources and sinks is justifiable even in much warmer climates, where vapor content would increase substantially above current values. They also make valiant attempts to reconcile their findings with earlier studies documenting the mass-sink effect in tropical cyclones and other mesoscale systems. In the end, they find that the source/sink effect is indeed justifiably neglected for midlatitude climate modeling applications, even in very warm climates. This result is well-supported, and the analysis is compelling. I appreciate the thorough analysis, and the complete exploration of the climate parameter space. Other than running with an SST that is too cold in the "TC-world" experiments, and some surprising results from the model, I have only a few major concerns.

Thank you for your comments. Including more information about precipitation rates in our simulations will significantly improve the paper, and we appreciate you pressing us on this point. We think, for reasons outlined below, that using a 300 K SST in our TC world simulations is not cause for concern (in brief, because the threshold for TC genesis goes up and down with climate warming or cooling), and have chosen to keep results from 300 K TC world simulations in the revised paper because they produce precipitation rates comparable to the simulations described by Lackmann and Yablonsky (2004). Detailed responses to all of your comments are below.

The results of the current paper are somewhat surprising given that precipitation rate increases would surely be significant in the warmer model simulations. That 25 mm of rainfall corresponds to 2.5 hPa of hydrostatic pressure mass is inescapable, which must mean that the mass loss is entirely compensated by horizontal convergence in these simulations. At rather coarse model grid spacing, geostrophic adjustment is expected to prevent complete compensation at such large scales relative to the Rossby radius. There is a feedback involving moisture convergence, but it must evidently not be very effective, given the results. But this leads to a suggestion:

- Other than Fig. 4c (which is surprising in itself), there are not any other plots of precipitation, which is perhaps the easiest variable for readers to use in evaluating realism. Can you please add or show additional plots showing precipitation distribution with temperature for non-extreme events?

Your point about additional plots of precipitation being useful for evaluating realism is a good one. We have added a figure (Fig. 3 in the revised manuscript, summarized on lines 161-163 in the text) showing snapshots of midlatitude precipitation fields for the simulations currently shown in Fig. 2 (all treatments of the mass sink in the $\alpha = 1$ and $\alpha = 6$ simulations). The precipitation fields for simulations at $\alpha = 1$ should be useful for evaluating realism in a climate state comparable to modern Earth, and the precipitation fields for simulations at $\alpha = 6$ should provide some information about how the intensity and spatial structure of precipitation changes in much warmer climates.

- More could be said about why extreme precipitation decreases in the warmest simulations. And on line 164, do you mean lower percentiles?

The decrease in "extreme" precipitation rates in Fig. 4c in the original manuscript appears because we calculate high-percentile precipitation rates for all times and not just times when rain occurs (a detail we have clarified on lines 185-186 in the revised manuscript), and 99th percentile precipitation rates are not sufficiently far into the tail of the precipitation distribution to show the consistent increases with warming expected of the most extreme precipitation rates. Instead, they behave more like the mean midlatitude rain intensity (i.e., the average midlatitude precipitation rate conditioned on the occurrence of rain), which also decreases in warm climates, likely due to decreases in the vigor of midlatitude eddies (see response to your next comment). The text on lines 163-164 in the original manuscript was correct: at higher percentiles, extreme precipitation rates increase across the entire temperature range. For the sake of a streamlined presentation, we have replaced Fig. 4c (Fig. 5c in the revised manuscript) with a plot of 99.9th percentile precipitation rates, which increase with warming across the entire temperature range as discussed on lines 193-198.

- The cyclones in Fig. 2 with alpha=6 look rather anemic, and I wonder if there are other changes (e.g., in the static stability) that are inhibiting their development? Could you please plot mid-latitude potential temperature profiles for the runs? What might be causing this?

Midlatitude eddy kinetic energy decreases in warm climates due to reductions in the meridional temperature gradient and increases in the midlatitude dry static stability, both of which reduce mean available potential energy (see O'Gorman and Schneider (2008)). We have added a more thorough summary of the results from O'Gorman and Schneider (2008), and in particular now note that reductions in midlatitude eddy kinetic energy can be linked to changes to the midlatitude temperature structure (lines 153-159). Because O'Gorman and Schneider (2008) already provides a detailed discussion of changes to the midlatitude temperature structure in similar experiments with an almost-identical model, we don't think that adding a plot of midlatitude potential temperature profiles is necessary to adequately address this comment.

For the 10Xms simulations, 25 mm of rainfall is 25 hPa pressure-mass reduction, which surely must be a powerful deepening mechanism for precipitating cyclones. Thus it is very

surprising that this doesn't increase EKE, but again, that depends on the model producing realistic rain rates in such cyclones, which are not fully shown. That the rain rate increases at a given temperature in the 10X simulation indicates increased convergence and moisture convergence, but this evidently doesn't produce appreciable additional vortex stretching.

We agree that it's surprising that including a 10x mass sink has little effect on EKE (though, as Fig. 4c and Fig. 5 show, it does have an effect on other fields), and we expect that the additional plots of precipitation fields (Fig. 3) will help to convince readers that this is not due to unrealistically-low precipitation rates in midlatitude cyclones. It's possible that large-scale constraints on EKE (imposed, for example, by the mean available potential energy) limit the extent to which it can change in response to the inclusion of strong mass sinks. The original manuscript contained some text speculating that this may help explain why EKE remains unchanged even in 10xMS simulations, and we made a small change (line 204) to better highlight the contrast between EKE and other statistics and our speculative explanation for the difference.

My other concern is related to the "TC-world" simulations, and this is why I recommend borderline major revisions. These experiments should be re-done, because those presented in section 5 were run with a constant SST of 300K (26.85C), which is marginal for TC genesis even in today's climate (traditionally, 26.5C is used as a cut-off value, though other studies have debunked this). The experiments are run in a marginal TC environment, so it is no wonder it was difficult to get them to develop at these grid lengths. Large regions of the tropical ocean feature SST above this value even in present-day conditions (30C or higher), and the results will be sensitive to this especially at coarse grid spacing. I suggest setting the SST to 305 and 310K, and you could see very different results. Perhaps also reduce the Earth radius further, if computational resources allow.

The threshold SST for TC genesis is climate-dependent, and the fact that it is around 26.5 C on Earth (where, as your comment mentions, SSTs above 30 C are fairly common) does not mean that it is also a cut-off value for TC genesis in our TC world simulations (where the SST is 26.85 C everywhere). Past work (e.g., Merlis et al., 2016) shows that TCs form readily in TC world simulations at SSTs as low as 280 K, far below the SST we use. We have added text on lines 435-438 noting, with a citation to Merlis et al. (2016), that the choice of SST is not crucial for TC genesis in TC world simulations, and that our choice of SST produces precipitation rates comparable to LY04's Eta model simulations.

Further reducing the planetary radius of TC world simulations does meaningfully increase computational cost (it requires a shorter time step, which increases the model run time by about a factor of 2 for each factor-of-2 reduction in radius), and quarter-Earth-radius simulations already require about a week of wall time each. It's moreover not clear that simulated TCs would fit on an eighth-Earth-radius planet, since TC size is already comparable to the planetary surface area in quarter-Earth-radius simulations (see Fig. 9c). We have added text on lines 467-471 acknowledging that higher-resolution simulations would be helpful for evaluating robustness and explaining why we think that quarter-Earth-radius simulations

are as far as we can push this model.

Related questions and suggestions:

- Again, what are the rain rates in these TCs? It is essential to see these values to assess whether these experiments are useful in reconciliation of the results with prior studies. Please share plots of rainfall for the TCs.

  Again, this is a good point. We have added plots of precipitation fields from TC world simulations to Fig. 9, using the same accumulation period as a key figure in LY04, and edited the text on lines lines 456-461 documenting the similarity between precipitation rates in our simulations and LY04's Eta model simulations.

- Is the increase in hyperdiffusion by a factor of 5-10 really necessary, and if not, can it be relaxed? Could this not be removing meaningful gradients and reducing the effect in question?

  It is not necessary, though it does help produce TCs with stronger central pressure anomalies. We ran an additional set of quarter-Earth-radius TC world simulations without the hyperdiffusion and found no evidence that including precipitation mass sinks produces systematically deeper storms (the effect that our work focuses on), though we did find some tentative evidence that mass sinks may help maintain weak TCs that would otherwise dissipate. We added a brief note about these simulations on lines 498-501 and document results in more detail in Appendix E.

The question of spatial scale remains. The authors do mention frontal scales (such as in the studies by Qui et al.), and the authors mention this around line 533. But for scales much smaller than the Rossby radius, we expect divergent flow to more readily remove perturbations resulting from the mass sink.

Agreed it is not clear that small-scale balanced flows should be more substantially affected by the mass sink. Nonetheless we do see transient effects on individual cyclones in section 5 and we have modified lines 599-600 to point this out. To give more physical context, we now mention on lines 43-45 in the introduction that the mass sink will lead to decreases in surface pressure, but that this is partially counteracted (to an extent that depends on length scale) by convergent flow which leads to vortex stretching.

In discussing planetary atmospheres around lines 495-500 or 510-515, please consider adding studies of the Martian general circulation, which find that sublimation and deposition of CO2 play a substantial role (see, e.g., Chow et al. 2019, JGR Planets, and references therein).

Chow, K.C., Xiao, J., Chan, K.L. and Wong, C.F., 2019. Flow associated with the condensation and sublimation of polar ice caps on Mars. Journal of Geophysical Research: Planets, 124(6), pp.1570-1580.

We added references to Chow et al. (2019) and several related papers around lines 578-581, where we now discuss the role of carbon dioxide condensation in the Martian general

circulation.

Some other useful references that present equations for moist atmospheres are Ooyama 2001, and Bott 2008, and references therein. It would be good to add these to the list and discuss how such processes may indeed become important at very small, non-hydrostatic grid lengths.

Bott, A., 2008. Theoretical considerations on the mass and energy consistent treatment of precipitation in cloudy atmospheres. Atmospheric research, 89(3), pp.262-269.

Ooyama, K.V., 2001. A dynamic and thermodynamic foundation for modeling the moist atmosphere with parameterized microphysics. Journal of the atmospheric sciences, 58(15), pp.2073-2102.

We added references to Ooyama (2001) and Bott (2008) around lines 604-606, where we now discuss the possibility that mass sinks may be more important on very small scales.

A few other specific questions:

Figure 2. For the alpha=6 simulations, the cyclones don't look well. How does the layer-average static stability change relative to the alpha=1 simulations? Is this affecting the results? (As mentioned above)

The midlatitude layer-average static stability increases with warming, as discussed in O'Gorman and Schneider (2008). This reduces the mean available potential energy, and contributes to a reduction in eddy kinetic energy. We added text on lines 152-161 discussing the effects of warming on mean available potential energy and eddy kinetic energy (relying on results from O'Gorman and Schneider (2008), as discussed earlier in this response), and we now mention that reductions in MAPE and EKE (possibly alongside a change to the dominant mode of midlatitude instability) may be why midlatitude cyclones look less vigorous in warm climates.

Figure 8. It is quite surprising that the near-surface wind speeds don't seem to increase with decreasing grid length. Can you include the numerical value of maximum wind-speed value in each simulation in the figure, caption, or in the associated discussion? Why do the wind speeds not increase with resolution, or is that just peculiar to this particular snapshot?

Global-maximum wind speeds are slightly higher in TC world simulations with larger radii (lower spatial resolution), but this is largely because those simulations produce more storms. (Maximum wind speed varies between storms, so the strength of the strongest storm is likely to increase if you sample a larger number of storms even if the intensity distribution you draw from is fixed.) The maximum wind speed of the median TC (a measure of the intensity of a typical TC) is higher in simulations with smaller grid lengths, as expected. We have added numerical values of median TC wind speeds to Fig. 9, and added text in the Fig. 9 caption and on lines 482-486 providing details about the median wind speed calculation and discussing how TC wind speed varies with resolution.

Figure 9. Panel (f) is mislabeled.

We have fixed the figure label.

Appendix A3. The need for a mass fixer is explained here, but it is not obvious (to me). If P ∼ E, why is this necessary, exactly? Where the reinstated mass is added within the model domain, or is it a sort of global distribution? Are you adding dry air mass in precisely the locations where the precipitation sink is removing it?

The mass fixer is required to maintain a fixed amount of dry (non-water) mass in the atmosphere. The dry atmospheric mass should be constant because there are no sources or sinks of dry air in our simulation, but numerical errors in our implementation of sources and sinks of moist mass produce unphysical reductions in dry mass. (This is shown by Eq. A12.) The mass fixer compensates for those unphysical reductions in dry mass and prevents a gradual downward drift of total atmospheric mass. The mass fixer adjusts the total atmospheric mass by multiplying surface pressure by a single number in all columns. Because our model uses sigma coordinates, this changes the mass thickness of each level by an amount proportional to the level's mass thickness. The model then divides specific humidity in each grid cell by the same (single) value, which ensures that the total water mass in the atmosphere does not change and that the mass fixer only adjusts the atmospheric dry mass. This procedure does not add dry mass in precisely the places where it is lost during the calculation of sources and sinks of water mass. We think that adding dry mass in precisely the locations where it is lost is unnecessary because changes in dry mass (proportional to $dq_v^2$) are locally small compared to changes in moist mass (proportional to $dq_v$), and the mass fixer is used primarily to allow stable long-term simulations that correctly conserve total dry mass. We have added text to emphasize why the mass fixer is necessary (lines 669-673), clarify where dry mass is added back to the atmosphere (lines 699-701), and explain why we don't add dry mass in precisely the locations where it is lost due to numerical errors (lines 701-704).

**RC 2**

**Synopsis**

This study deals with the influence of precipitation mass sinks on the dynamics of large-scale weather systems using the GFDL atmospheric global circulation model. First, this effect is investigated in idealized aquaplanet simulations for different climate states. Mass sinks proved to have little impact on the statistics of weather systems but an artificial tenfold amplification of the mass sinks leads to notable effects. A proper analysis of the potential vorticity source and sinks provide an explanation for this result. Furthermore, the influence of mass sinks in simulations of tropical cyclones (TCs) was investigated. For this purpose the surface of the aquaplanet has been prescribed by 300K and the Coriolis parameter was set to a constant which yields a planet covered with many TCs (TC world). While the mass sink has also in these simulations nearly no impact on the long term TC statistics, it can alter the evolution over times scales on 1-3 days.

**Comments and recommendation**

Although this study has been conducted properly, I don't think that it provides new insight into atmospheric dynamics.

Thank you for your comments. They highlight an important shortcoming in the original manuscript; namely, a lack of discussion of our choice to use a pseudoadiabatic model and of the possible consequences of that decision. For the reasons described below, we do not agree that the use of a pseudoadiabatic model means that this study provides no new insight into atmospheric dynamics—we think that it does—but we take responsibility for not adequately justifying our choice of model. We have made several changes (outlined below) to address the specific concerns brought up in your comments, and think that the changes significantly improve how we frame and discuss our results.

This conclusion is based on the following statements:

- The authors based their investigations on the claim that a mass sink exist within the interior of the atmosphere. They derived the according mass sink term in Appendix A1. This term is added to the mass continuity equation and it is obvious that they assume with this approach that water mass is annihilated. Although the pseudoadiabatic scheme of the model give rise to this assumption, it is not possible. Instead, water remains in liquid or frozen form in the atmosphere. The only mass sink appears at the surface where water leaves the atmosphere via precipitation.

  As you say, our derivation of the mass sink term in Appendix A1 relies on the pseudoadiabatic assumption, and assumes that water mass disappears immediately upon condensation. Strictly speaking, this is not possible, but the same result can be obtained by deriving a mass sink term without assuming that water disappears immediately upon condensation, and then assuming that time scales for precipitation formation and fallout are infinitely fast. Because our study focuses primarily in the dynamics of midlatitude and tropical cyclones (large-scale weather systems that evolve over time scales of days), we think that treating precipitation formation and fallout as infinitely fast is a reasonable simplifying assumption, and not one that should preclude our study from providing new physical insight. We have added text on lines 123-138 justifying and discussing our use of a pseudoadiabatic model, and a derivation in Appendix B showing how the pseudoadiabatic version of the continuity equation follows from assuming infinitely fast precipitation formation and fallout.

  The sedimenting hydrometeors exert a drag force on the air that is identical to their weight which is known a condensate loading. Therefore, Eq. A2 cannot be true since liquid or frozen hydrometeors do not contribute to the density in this equation. That this matters has been shown, e.g. by Xu and Emanuel (1989).

  Equation A2 is simply a restatement of hydrostatic balance, and makes no assumptions about whether condensed water species contribute to the density and hydrostatic pressure fields. In Equation A1, the equality $\rho = (1 + r_v)\rho_d$ does explicitly neglect condensate loading. However, as described above, the same mass sink term can be

derived by using a version of Equation A1 that includes condensate loading, and later assuming that time scales for precipitation formation and fallout are infinitely fast (see Appendix B in the revised manuscript). More generally, the neglect of condensate loading in our study follows from our use of the pseudoadiabatic approximation, which we think is justifiable (see lines 123-138 in the revised manuscript). While we do not think that there is a clear reason to assume that the neglect of condensate loading has a first-order impact on this study's results, we agree that it's an important caveat to highlight, and we discuss it in greater detail in the revised paper (again, see lines 123-138).

The potential vorticity calculations in sections 4.1-4.3 are based on the incorrect mass sink term and are, therefore, obsolete. The mechanical and thermodynamical interaction of sedimenting hydrometeors with air leads to sources in the potential vorticity equation but I don't believe that they have the simple form as described in section 4.1.

The potential vorticity calculations in Section 4 are consistent with the mass sink term as it appears in our model, and so are credible to the extent that the mass sink term in our model is credible. As discussed above, we think that the version of the mass sink term used in our model, while approximate, is justifiable. It is true that the potential vorticity source from mass sinks takes a different form when prognostic hydrometeor fallout is considered, in which case it involves the divergence of the hydrometeor sedimentation mass flux (see e.g. Equation 20a in Schubert et al. (2001)). However, there is a connection between this version of the PV source term and the version used in our paper: if precipitation formation and fallout are assumed to be infinitely fast, then the divergence of the hydrometeor sedimentation mass flux can be replaced by the condensation rate, which gives the PV source term that appears in Equation 6. We added an appendix (Appendix D) showing that the the assumption of infinitely fast precipitation formation and fallout connects the pseudoadiabatic PV equation to a version of the PV similar the the one presented in Schubert et al. (2001) (except in potential temperature coordinates), and reference that appendix at the start of section 4 (lines 232-239) to ensure readers are aware of how the two versions of the PV equation are connected as well as how they differ.

To tackle the issue of the mass sink impact one should use a model that includes a cloud microphysical scheme and prognostic equations for liquid and frozen water. The simplified GFDL model used by the authors is likely not appropriate for this purpose. Consequently, I do not recommend publishing this article.

To summarize our response: it is certainly possible that effects our model neglects (e.g., prognostic hydrometeor fallout and condensate loading) could alter our results. We clearly acknowledge this in the revised paper (lines 133-135), and emphasize (lines 137-138 and 606-609) that we would welcome follow-up studies examining whether these effects enhance the dynamical impact of precipitation mass sinks. In our view, however, it's far from clear that a full cloud microphysics scheme and prognostic equations for

condensed water species are essential for our study, which focuses on the existence or absence of the mass sink itself. They could be details that have little impact on our results (for example, by changing the detailed vertical structure of PV anomalies from mass sinks without changing their overall magnitude), and it seems premature to conclude that a model that neglects them can provide no useful insight into the effects of precipitation mass sinks.

**References**

Bott, A. (2008). Theoretical considerations on the mass and energy consistent treatment of precipitation in cloudy atmospheres. *Atmospheric Research*, 89(3):262–269.

Chow, K.-C., Xiao, J., Chan, K. L., and Wong, C.-F. (2019). Flow associated with the condensation and sublimation of polar ice caps on mars. *Journal of Geophysical Research: Planets*, 124(6):1570–1580.

Merlis, T. M., Zhou, W., Held, I. M., and Zhao, M. (2016). Surface temperature dependence of tropical cyclone-permitting simulations in a spherical model with uniform thermal forcing. *Geophysical Research Letters*, 43(6):2859–2865.

Ooyama, K. V. (2001). A dynamic and thermodynamic foundation for modeling the moist atmosphere with parameterized microphysics. *Journal of the Atmospheric Sciences*, 58(15):2073–2102.

O'Gorman, P. A. and Schneider, T. (2008). Energy of midlatitude transient eddies in idealized simulations of changed climates. *Journal of Climate*, 21(22):5797–5806.

Schubert, W. H., Hausman, S. A., Garcia, M., Ooyama, K. V., and Kuo, H.-C. (2001). Potential vorticity in a moist atmosphere. *Journal of the Atmospheric Sciences*, 58(21):3148–3157.

---

## Author Response (AR2)

Note: this document contains co-editor and reviewer comments in black text and our responses in blue. Line numbers in our responses refer to the tracked-changes version of the manuscript.

**Co-editor**

Dear Dr. Abbott

Many thanks for your revisions and for addressing the points raised by the reviewers. Both reviewers are agree that the paper has further improved and is almost ready for publication. I therefore accept your paper for publication in WCD subject to a few minor corrections:

Reviewer 2 recommends adding "in the pseudoadiabatic limit" to the title, and I think this is a good idea. It makes the title more technical but also more specific about an important assumption of the study.

See response to Reviewer 2 below.

Reviewer 2 thinks that Appendices B and D are not required. However, I appreciated these details; please re-consider whether you like to keep them or not. And have a look at the consistency of notation: in L241 and L775 v is the horizontal velocity, why then not also use v in eq. (A4, A5, B3)?

See response to Reviewer 2 below. We've modified notation in Equations A4-A7 and B3 and on lines 628 and 631 so that $\mathbf{v}$ is consistently used for horizontal velocity.

Reviewer 1 has several recommendations concerning the clarity of the figures.

See response to Reviewer 1 below.

References: Copernicus journals use journal abbreviations and include a DOI. Please check in published WCD papers and adapt your list of references accordingly.

We have modified our list of references to match the style of Copernicus journals.

I am looking forward to receiving the final version of your manuscript.

With best regards,

Heini Wernli

**Reviewer 1**

The authors have addressed all of my earlier concerns. This will be a useful contribution, and I find the results both interesting and somewhat surprising. I do have a minor suggestion regarding the graphics. Several of the figures (1, 4, 5, and 10, for example) would benefit from grid lines.

We agree that grid lines would help readers estimate the coordinates of points shown in figures 1, 4, 5, and 10, but are concerned that they would add visual clutter to plots that already contain several different lines. As a compromise, we have added tick marks to the right and top panel edges in Figures 1, 4, 5, 10, A1, and E1, which will make it easier to estimate the coordinates of points that lie far from labeled axes.

In Figure 10, the lines are thick, and of similar color, making comparison difficult. Also, some of the ordinate labels (e.g., Panel c) seem missing or irregular.

We decreased the line thickness in Figures 10 and E1, and replaced dark blue curves with a lighter blue throughout the manuscript. The ordinate labels differ between panels in Figure 10 because the y-axis range changes; we modified ordinate labels so that they're identical in panels a, c, and e.

**Reviewer 2**

The authors considered all of my comments in their revised manuscript. Therefore, the paper has improved in my view. However, I am still not sure that the presented results describe the true impact of precipitation mass sinks on midlatitude storms. This study suggests that the magnitude of this impact is rather small. Therefore, it is not obvious why condensate loading has not an impact of similar magnitude and that the superposition of both impacts lead to a different result. Consequently, I recommend adding "in the pseudoadiabatic limit" to the title. Then the reader sees that the results may not describe the true impact and further work may be needed.

We agree that it's worthwhile to highlight the idealized nature of our simulations in the title, but worry that the meaning of "in the pseudoadiabatic limit" will be unclear without accompanying text clarifying that this refers to a limit where condensate immediately falls out of the atmosphere. Instead, we have changed the title to "Impact of Precipitation Mass Sinks on Midlatitude Storms in Idealized GCM Simulations over a Wide Range of Climates", and modified a sentence in the abstract to highlight that one of the assumptions made by this GCM is that condensate immediately falls out of the atmosphere (lines 7-8).

Appendix B and appendix D: In my view it is not necessary to show that the pseudoadiabatic model results from the limit of an infinitely fast fallout as it is obviously consistent with the statement of an immediate liquid water annihilation.

After some consideration we decided to keep these two appendices. Because they're appendices they shouldn't be distracting to readers who find them uneccecessary or obvious, and some readers may find them useful.

---

## Author Response (AR3)

We modified the colors used in Figures 1, 4, 5, and A1 so that they can be distinguished by readers with color vision deficiencies.